# In vitro maturation of *Toxoplasma gondii* bradyzoites in human myotubes and their metabolomic characterization

Céline Christiansen[1], Deborah Maus[1], Ellen Hoppenz[1], Mateo Murillo-León [2,3,4], Tobias Hoffmann [5], Jana Scholz [1], Florian Melerowicz [1], Tobias Steinfeldt [2,3], Frank Seeber[6] & Martin Blume [1✉]

The apicomplexan parasite *Toxoplasma gondii* forms bradyzoite-containing tissue cysts that cause chronic and drug-tolerant infections. However, current in vitro models do not allow long-term culture of these cysts to maturity. Here, we developed a human myotube-based in vitro culture model of functionally mature tissue cysts that are orally infectious to mice and tolerate exposure to a range of antibiotics and temperature stresses. Metabolomic characterization of purified cysts reveals global changes that comprise increased levels of amino acids and decreased abundance of nucleobase- and tricarboxylic acid cycle-associated metabolites. In contrast to fast replicating tachyzoite forms of *T. gondii* these tissue cysts tolerate exposure to the aconitase inhibitor sodium fluoroacetate. Direct access to persistent stages of *T. gondii* under defined cell culture conditions will be essential for the dissection of functionally important host-parasite interactions and drug evasion mechanisms. It will also facilitate the identification of new strategies for therapeutic intervention.

[1] NG2: Metabolism of Microbial Pathogens, Robert Koch-Institute, 13353 Berlin, Germany. [2] Institute of Virology, Medical Center University of Freiburg, 79104 Freiburg, Germany. [3] Faculty of Medicine, University of Freiburg, 79104 Freiburg, Germany. [4] Faculty of Biology, University of Freiburg, 79104 Freiburg, Germany. [5] ZBS 4: Advanced Light and Electron Microscopy, Centre for Biological Threats and Special Pathogens 4, Robert Koch-Institute, 13353 Berlin, Germany. [6] FG 16: Mycotic and Parasitic Agents and Mycobacteria, Robert Koch-Institute, 13353 Berlin, Germany. ✉email: blumem@rki.de

*T*oxoplasma gondii is an apicomplexan parasite that infects most warm-blooded animals, including an estimated third of all humans[1]. Similar to other infectious protozoa, including the apicomplexan *Plasmodium vivax*, the kinetoplastidae *Trypanosoma cruzi* and *Leishmania* spp., *T. gondii* forms specialized persistent stages that resist immune responses and many medical treatments[2]. *T. gondii* bradyzoites reside, surrounded by a glycan-rich cyst wall, predominantly in brain and muscle tissue and can be transmitted through ingestion of undercooked meat products[3]. These largely asymptomatic infections can cause recurring disease in immune-weakened individuals that are lethal if untreated[4].

The rapid proliferation of tachyzoites and the slow replication or dormancy of bradyzoites exert distinct demands on the metabolism of *T. gondii*. The rate of proliferation and the metabolic state are closely interrelated. They have been shown to independently affect lethality of antibiotics against bacterial microbes, with metabolism being the dominant factor[5]. While multiple metabolomic investigations on the tachyzoite stage of *T. gondii* revealed unexpected features of key metabolic pathways such as the GABA-shunt of the tricarboxylic acid (TCA) cycle, a constitutively active gluconeogenic fructose 1,6-bisphosphatase, and the presence of a non-oxidative pentose phosphate pathway[6–8], the metabolism of bradyzoites has only been characterized indirectly[9], largely due to experimental constraints. In contrast to tachyzoites, bradyzoites depend on the turnover of the storage polysaccharide amylopectin by glycogen phosphorylase[10], the hexose kinase[11] and on a dedicated isoform of lactate dehydrogenase[12]. This indicates that glycolytic degradation of glucose plays an important role in bradyzoites. Similarly, the relevance of mitochondrial and amino acid metabolism has been inferred. Bradyzoite formation is induced by a number of electron transport inhibitors, including atovaquone, rotenone, oligomycin and antimycin[13,14]. Moreover, *T. gondii* cysts survive extended atovaquone exposure in vivo[15–17], indicating that their mitochondrial electron transport chain (mETC) is not strictly essential. Similarly, bradyzoites can also be induced by limiting supply of exogenous amino acids[14], and their viability depends on proteolysis in their plant-like vacuolar compartment[18] and on autophagy[19]. Collectively, these data indicate a substantial remodeling of metabolic homeostasis in bradyzoites as a strategy to cope with nutrient limitations[20]. This proposed metabolic shift may also be associated with tolerance against many antiparasitic treatments, as the metabolic state of microbes has important implications for the lethality of antimicrobials[5].

However, our ability to investigate such mechanisms remains very limited by the lack of adequate in vitro culture models[2].

Current methods to experimentally generate mature *T. gondii* tissue cysts are confined to murine infections with cystogenic *T. gondii* strains[21,22]. The number of obtainable cysts from this in vivo model is low, precluding their metabolic analysis with current mass spectrometry-based approaches. In contrast, in vitro models are scalable and the stage conversion of *T. gondii* can be induced by a variety of treatments, such as alkaline, heat and chemical stress[23], nutrient starvation[24], excess adenosine[25], inhibitors of parasite protein kinases[26] as well as host cell-dependent factors such as cell cycle arrest and low glycolytic flux[20,27,28]. In addition, the infection of C2C12 murine skeletal muscle cells, neurons[29,30], and primary murine brain cells[31] facilitates spontaneous stage conversion of cystogenic strains. Generally, maturation times of in vitro cysts remain brief, but cysts develop detectable tolerance against short exposure to low doses of pyrimethamine after three days[10] and become orally infectious to mice at high doses after five days of culture[32]. However, establishment of long-term cultures that allow development of pan-antimicrobial- and temperature stress-tolerant cysts[33,34] remains a challenge. Incomplete differentiation leads to expansion of proliferating parasite populations and subsequent host cell lysis that limits the scale and duration of current in vitro culture systems.

Here, we developed a human myotube-based in vitro culture system that is scalable and enables long-term maturation of *T. gondii* cysts. The cysts resemble in vivo cysts in their ultrastructure, tolerance to antiparasitics and temperature stress and infectiousness to mice. Mass spectrometry-based metabolic profiling revealed a distinct metabolome that renders tissue cysts insensitive toward the aconitase inhibitor sodium fluoroacetate (NaFAc).

## Results

**KD3 human myoblasts differentiate into multinucleated myotubes.** *T. gondii* persists in skeletal muscle tissue[35], but robust in vitro culture systems supporting long-term maturation of tissue cysts in these natural host cells are lacking. We tested an immortalized human skeletal muscle cell isolate (KD3)[36] for its ability to support the culture of *T. gondii* tissue cysts. Subconfluent myoblast cultures were differentiated into myotubes through serum starvation for five days until cells fused to multinucleated tubes that were expressing the myosin heavy chain protein (Fig. 1A) and developed spontaneous contraction activity (Supplementary Movie 1, 2)[36]. Accordingly, the myogenic index, which reflects the fraction of nuclei residing in cells containing three or more nuclei, rose to 0.3 (Fig. 1B).

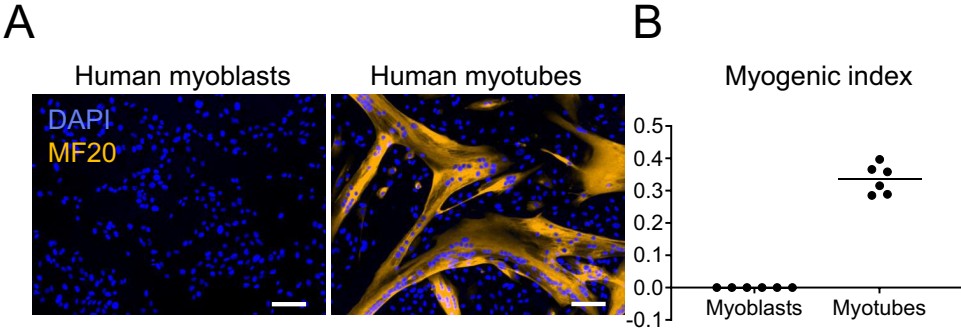

**Fig. 1 KD3 human myoblasts form multinucleated myotubes. A** Immunofluorescence images of KD3 myoblasts and myotubes after 5 days of differentiation, stained for myosin heavy chain (MF20) and DNA (DAPI). Representative of at least two independent experiments. Scale bar indicates 100 μm. **B** As a morphological parameter of myotube formation, the myogenic index (number of nuclei residing in cells containing three or more nuclei, divided by the total number of nuclei) was measured in at least ten different randomly selected locations by CCEAN software[93]. Values are expressed as mean of two individual experiments performed in triplicates. Source data are provided as a Source Data file.

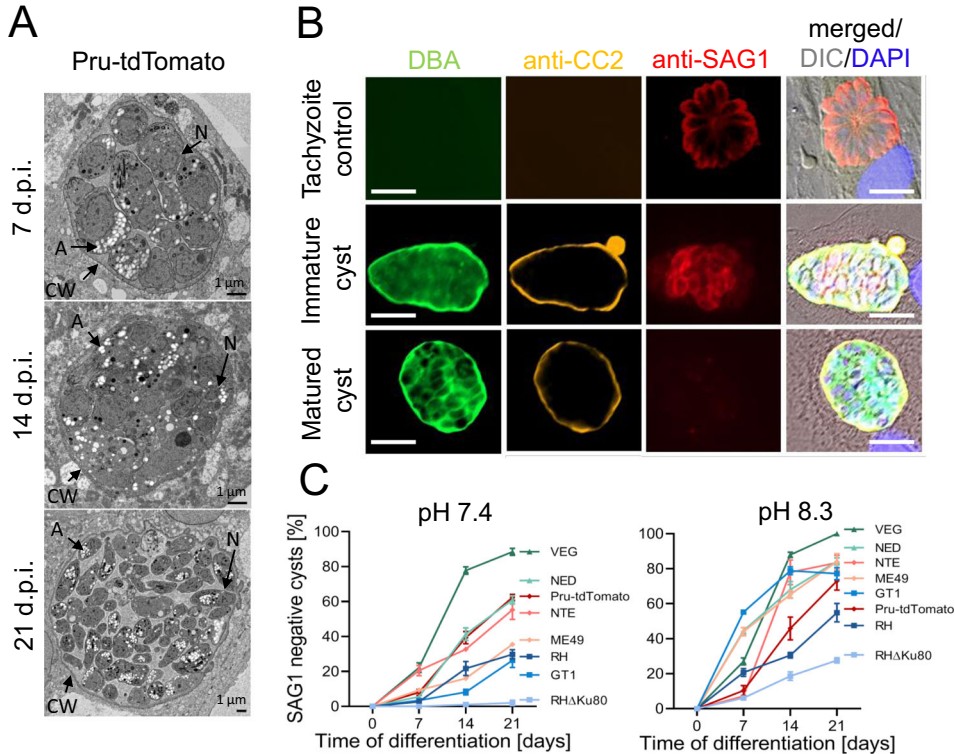

**Fig. 2 Time- and pH-dependent cyst maturation of type I, II, and III *T. gondii* strains in KD3 myotubes. A** Electron microscopy of 7-, 14- and 21-day-old Pru-tdTomato tissue cysts in KD3 myotubes (days post infection; d.p.i.). Cysts show a distinct wall (CW) and are densely packed with parasites containing amylopectin granules (A) and nuclei (N). Representative of at least two independent experiments. **B** Immunofluorescence imaging of KD3 myotubes infected with type I (RH, RHΔ*ku80*, GT1), type II (ME49, NTE, Pru-tdTomato) and type III (NED, VEG) parasites. Cyst formation was induced for 7, 14, and 21 days under neutral or basic pH and stained with anti-CC2, anti-SAG1 antibodies, DBA and DAPI. Shown are representative images of data shown in (**C**) of tachyzoite controls infected with RHΔ*ku80* for 24 h (upper panel), 14-day-old intermediate cysts (middle panel) and mature cysts (lower panel) of the NED strain. Scale bar indicates 5 μm. **C** Relative numbers of SAG1-negative but DBA- and CC2-positive cysts counted from three independent blinded experiments performed in triplicates. At least 25 DBA-positive cysts per strain, time point and replicate, respectively, were counted. Values represent means ± SEM. Source data are provided as a Source Data file.

**KD3 human myotubes support development of matured tissue cysts of several *T. gondii* strains.** Next, KD3 myotubes were tested for their ability to support long-term maturation of *T. gondii* cysts. We infected myotubes with Pru-tdTomato parasites, a type II Prugniaud strain constitutively expressing the tandem (td) Tomato variant of the red fluorescent protein, for up to 21 days. Evaluation of the ultrastructure of the encysted parasites by electron microscopy (Figs. 2A, S1) revealed hallmarks of bradyzoites, including abundant amylopectin granules to posteriorly positioned nuclei after seven days of infection. The formation of the cyst wall is indicated by the apparent deposition of electron-dense vesicular-tubular material along the parasitophorous vacuolar membrane throughout the course of infection (Fig. S1D)[37].

*T. gondii* strains differ greatly in their capacity to form tissue cysts in vitro and in vivo[38] and their maturation is accompanied by sequential regulation of marker protein expression[39]. To test the suitability of KD3 myotubes to culture tissue cysts of multiple *T. gondii* strains, we monitored the formation of tissue cysts of eight parasite strains of three major isotypes under $CO_2$-deplete conditions at neutral and basic pH. Stage conversion was detected by staining the cyst wall with *Dolichos biflorus* agglutinin (DBA) and antibodies against the CC2 protein[40] and by the absence of the tachyzoite-specific surface antigen 1 (SAG1) protein (Fig. 2B). Myotubes infected with type I RHΔ*ku80* parasites that do not easily form tissue cysts in vitro served as a tachyzoite control.

Strikingly, under both conditions all observed vacuoles of every tested strain were DBA-positive and remained stable without being overgrown by tachyzoites for 21 days. This was also true for the hard to differentiate RHΔ*ku80* (Fig. S2). We observed almost complete differentiation of type III VEG parasites into DBA- and CC2-positive and SAG1-negative cysts. All type II strains and the type III NED strain showed intermediate maturation (between 35 and 63%) that was considerably increased in basic pH to 73% and 83%, respectively. Also, wild-type type I strains GT1 and RH exhibited differentiation up to 30% that increased to 77% and 55% at basic pH, respectively. As expected, the high-passage laboratory strain RHΔ*ku80* did not loose SAG1 signal at neutral pH but required basic conditions (Fig. 2C).

To directly relate the capability of KD3 myotubes to generate mature bradyzoites to existing culture methods, we differentiated Pru-tdTomato parasites for up to 21 days in HFF, KD3 myoblasts and KD3 myotubes by bicarbonate starvation at pH 7.4 (Fig. S3A) and pH 8.3 (Fig. S3B). In both conditions, parasites in myoblasts and HFF cells lysed host cells after 14 and 21 days respectively. Only in myotubes, Pru-tdTomato parasites formed an increasing fraction of DBA-positive and SAG1-negative cysts, while SAG1-positive and DBA-negative tachyzoites remained undetectable.

We also tested spontaneous stage conversion under standard bicarbonate-replete conditions as observed in the murine skeletal muscle cell line C2C12[41]. To this end, we infected myoblasts, myotubes and HFF cells with Pru-GFP parasites which express GFP through the bradyzoite-specific *LDH2* promoter[42] for 96 h (Fig. S4). GFP-negative tachyzoites continued to proliferate in HFF cells and myoblasts leading to host cell lysis. In contrast, proliferation of *T. gondii* appeared

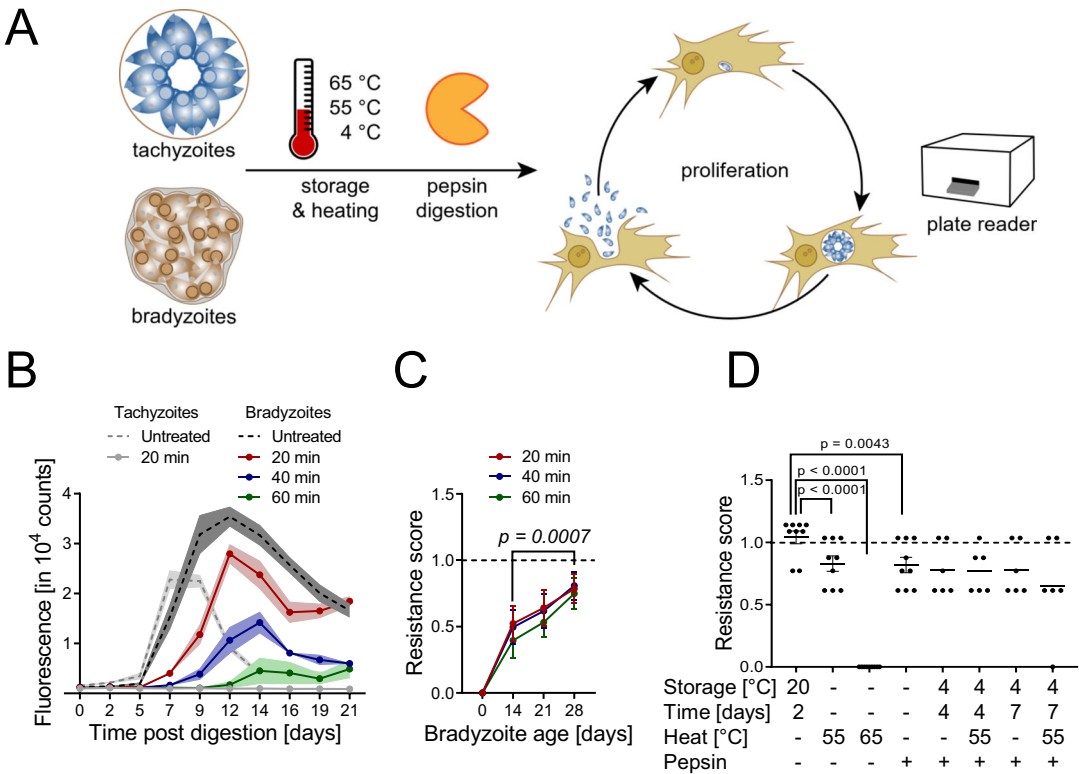

**Fig. 3 In vitro bradyzoites develop resistance to pepsin and tolerance to temperature stresses. A** Experimental design of in vitro pepsin and oral transmission assay. Pru-tdTomato cysts and tachyzoite controls were challenged with indicated pepsin and/or temperature stresses. Treated parasites were seeded onto fresh human fibroblasts to induce tachyzoite re-differentiation and proliferation. tdTomato fluorescence was monitored in a fluorescent based plate reader. **B** Raw fluorescence intensities of 21-day-old cysts digested for 20 (red), 40 (blue) and 60 min (green) and the respective tachyzoite control (gray). Dashed lines indicate corresponding untreated controls. The data represent the means and SEM of one experiment consisting of three digestion reactions read out in triplicate cultures. Undigested tachyzoites data represent two mock digestions. **C** Resistance scores of tachyzoite cultures (0), 14-, 21-, and 28-day-old cysts after 20 (red), 40 (blue), 60 min (green) of pepsin digestion. Shown are the means and SEM of three independent experiments performed in triplicates ($p = 0.0007$, two-tailed Mann–Whitney $U$ test on pooled 14 and 28-day scores). **D** Resistance scores of 35-day-old bradyzoites exposed to indicated temperature stresses and pepsin digestion. Values are expressed as means and SEM from three or two independent experiments in triplicates $p < 0.0001$ or $p = 0.0043$ as indicated, two-tailed Mann–Whitney $U$ test. Source data are provided as a Source Data file.

attenuated and spontaneous formation was indicated by DBA staining and GFP expression (Fig. S4).

Together, these data show that KD3 myotubes enable the culture of tissue cysts of multiple parasite strains at physiological pH over the course of 21 days and augment spontaneous stage conversion.

**KD3 myotubes-derived tissue cysts harbor pepsin- and temperature stress-resistant bradyzoites.** To test whether in vitro bradyzoites also develop functional hallmarks of in vivo bradyzoites, such as resistance to temperature stress and pepsin digestion[33], we infected myotubes with Pru-tdTomato parasites and then differentiated parasites for up to 35 days (Fig. 3A). Intracellular cysts were digested with pepsin for 20, 40, and 60 min, the reaction mix was then neutralized and cysts were seeded onto HFF cells to allow re-differentiation in tachyzoite medium under $CO_2$-replete conditions. Untreated and pepsin-digested tachyzoite cultures served as positive and negative controls, respectively. Growth of both, untreated tachyzoite and bradyzoite cultures, was observed by an increase in red fluorescence (indicative of tachyzoite replication) after two and seven days, respectively (Fig. 3B). Interestingly, all pepsin-treated bradyzoite cultures recovered after 7, 9, and 14 days, depending on the length of the pepsin digest, while pepsin-digested tachyzoite

cultures did not recover even after a period of 21 days. We quantified pepsin resistance across the time course of cyst maturation by calculating a resistance score (RS) (see "Methods") based on the pepsin-inflicted growth delay (Fig. 3C). Already after 14 days all digested cultures recovered with an RS between 0.4 and 0.5. Pepsin resistance continued to increase over the course of 28 days up to an RS of 0.8 (Fig. 3C).

In human transmission scenarios, encysted bradyzoites survive extended cold-storage and heat stress[33]. To test whether in vitro bradyzoites would survive under these conditions, cysts were matured for 35 days and treated with a combination of (1) storage as cell 'pellets' at 4 °C for up to seven days, (2) a 5 min heat shock at 55 °C or 65 °C and (3) pepsin digestion for 60 min (Fig. 3D). Storage at room temperature (RT) for two days served as a positive control and did not decrease viability in comparison with untreated cysts, as indicated by an RS of 1. Different combinations of storage at 4 °C for four or seven days and heat shock at 55 °C did lower the RS to a range between 0.75 and 0.65, while heating to 65 °C effectively killed tissue cysts as expected[33]. These results demonstrate that myotube-grown encysted bradyzoites develop traits that are important for oral transmission of tissue cysts as it occurs with cysts in muscle tissues of farm animals.

Summarized, these data indicate that KD3 myotubes support maturation of *T. gondii* tissue cysts for 35 days. These in vitro bradyzoites develop resistance to pepsin and temperature stresses,

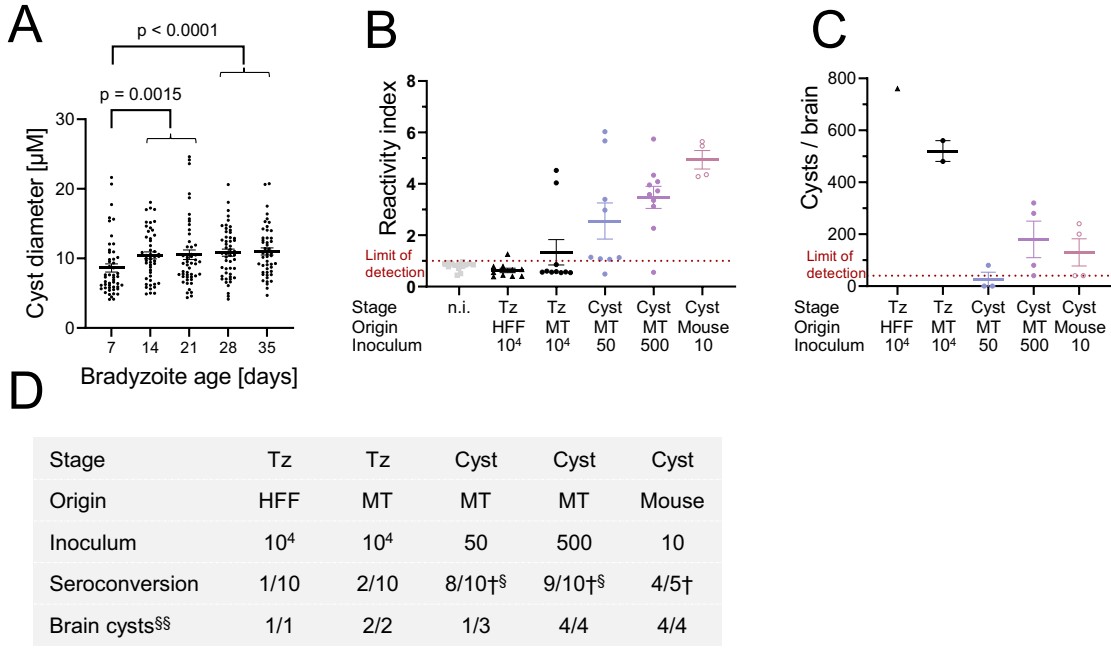

| Stage | Tz | Tz | Cyst | Cyst | Cyst |
|---|---|---|---|---|---|
| Origin | HFF | MT | MT | MT | Mouse |
| Inoculum | $10^4$ | $10^4$ | 50 | 500 | 10 |
| Seroconversion | 1/10 | 2/10 | 8/10†§ | 9/10†§ | 4/5† |
| Brain cysts§§ | 1/1 | 2/2 | 1/3 | 4/4 | 4/4 |

**Fig. 4 In vitro cysts are orally infectious to mice and lead to manifestation of chronic infection. A** Quantification of cyst diameter of 7-, 14-, 21, 28, and 35-day-old Pru-tdTomato in vitro tissue cysts. Encysted bradyzoites were stained with DBA and DAPI. Images of 50 cysts were randomly recorded and the diameter was measured by defining the DBA stain as the outline of the cysts. Values are expressed as means and SEM from one experiment. $p = 0.0015$ and $p < 0.0001$ two-tailed Mann–Whitney $U$ test **B–D** Reactivity index and cyst number per brain of mice that were orally infected with $10^4$ HFF- or myotube-derived tachyzoites, 50 or 500 35-day-old Pru-tdTomato in vitro cysts or 10 Pru-tdTomato in vivo cysts. Brains and blood from moribund animals were collected starting from 12 days post infection. **B** Seroconversion was checked for *T. gondii*-specific IgGs via ELISA and reactivity index was calculated. Values are expressed as means and SEM from two independent experiments consisting of five mice per group. **C** Quantification of cyst burden in brains of infected mice. Only seropositive mice from one experiment were analyzed for brain cysts. Values are expressed as means and SEM. Colored symbols are matched between identical experimental groups in **C** and **D**. **D** Summary of in vivo infection experiments. †: infected animals died during the 30-day period. § Seroconversion of moribund, but alive animals were checked throughout the experiment. §§ Brain cyst burden was only determined in seropositive animals of one experiment. Source data are provided as a Source Data file.

which are functional hallmarks of mature in vivo bradyzoites and required properties for both experimental oral infection of mice and environmental transmission to humans.

**KD3 myotube-derived tissue cysts are orally infectious to mice.** To assure oral infectivity to animals, groups of five mice were infected with 50 and 500 35-day-old in vitro cysts via oral gavage. Control groups included mice receiving $10^4$ fibroblast- or myotube-derived tachyzoites and as positive control 10 in vivo tissue cysts prepared 35 days post infection from brain tissue. To facilitate the comparison of infectivity between in vitro and in vivo cysts, we estimated relative volumes of in vitro cysts by measuring their diameters (Fig. 4A). Maturation over the course of 35 days increased the average diameter from 8.8 to 11 µm. This compares to 35-day-old in vivo cysts from brain tissue with a mean diameter of 56 µm[22]. Assuming spherical cysts, this implies an approximate 150-fold difference in volume. Infection of mice with 50 and 500 in vitro-derived cysts led to seroconversion of 8 out of 10 and 9 out of 10 mice, respectively (Fig. 4B, D). We also observed seroconversion of 1 and 2 mice infected with HFF- and myotube-derived tachyzoites. As expected, all mice infected with 10 in vivo-derived tissue cysts showed seroconversion. Infection was also indicated by weight loss of mice receiving 500 in vitro or 10 in vivo cysts. In these groups two animals succumbed to the infections while one animal that received 50 cysts needed to be culled (Fig. S6A, B). The presence of DBA-positive *T. gondii* cysts in the brains of infected mice was quantified microscopically 30 days post infection in brain homogenates (Fig. 4C, D). We

detected cysts in all seroconverted tachyzoite-infected mice and mice receiving 500 in vitro cysts or 10 in vivo cysts. Only one of three mice that seroconverted from infection with 50 cysts and survived, harbored detectable cysts.

Together these data indicate that KD3 myotube-derived tissue cysts develop oral infectivity to mice that is similar to in vivo cysts.

**KD3 myotube-derived tissue cysts develop resistance against antiparasitics.** *T. gondii* tissue cysts largely resist current toxoplasmosis treatments. To test whether cysts cultured in myotubes mirror drug resistance in vivo, we exposed 14-, 21-, and 28-day-old Pru-tdTomato-expressing cysts to antiparasitics for seven days and monitored their re-differentiation into tachyzoites under $CO_2$-replete conditions as a measure of bradyzoite viability (Fig. 5A). Figure 5B shows the fluorescence signal observed over 49 days, including 14 days of cyst maturation, seven days of pyrimethamine treatment, and 28 days of re-differentiation. Growth of DMSO- and pyrimethamine-treated parasites was indicated by an increase in fluorescence above background at 14 and 24 days post treatment, respectively (Fig. 5B). Next, 14-, 21-, and 28-day-old tissue cysts were treated with 5 µM and 20 µM pyrimethamine, 20 µM sulfadiazine and 0.1 µM of 1-hydroxy-2-dodecyl-4(1)-quinolon (HDQ), an inhibitor of dihydroorotate dehydrogenase (DHOD)[43] and alternative NADH dehydrogenase[44] (Fig. 5C). Bradyzoite cultures survived all tested treatments. Interestingly, while resistance against sulfadiazine, HDQ or low doses of pyrimethamine was already fully developed after 21 days, indicated by an RS of 1, resistance against

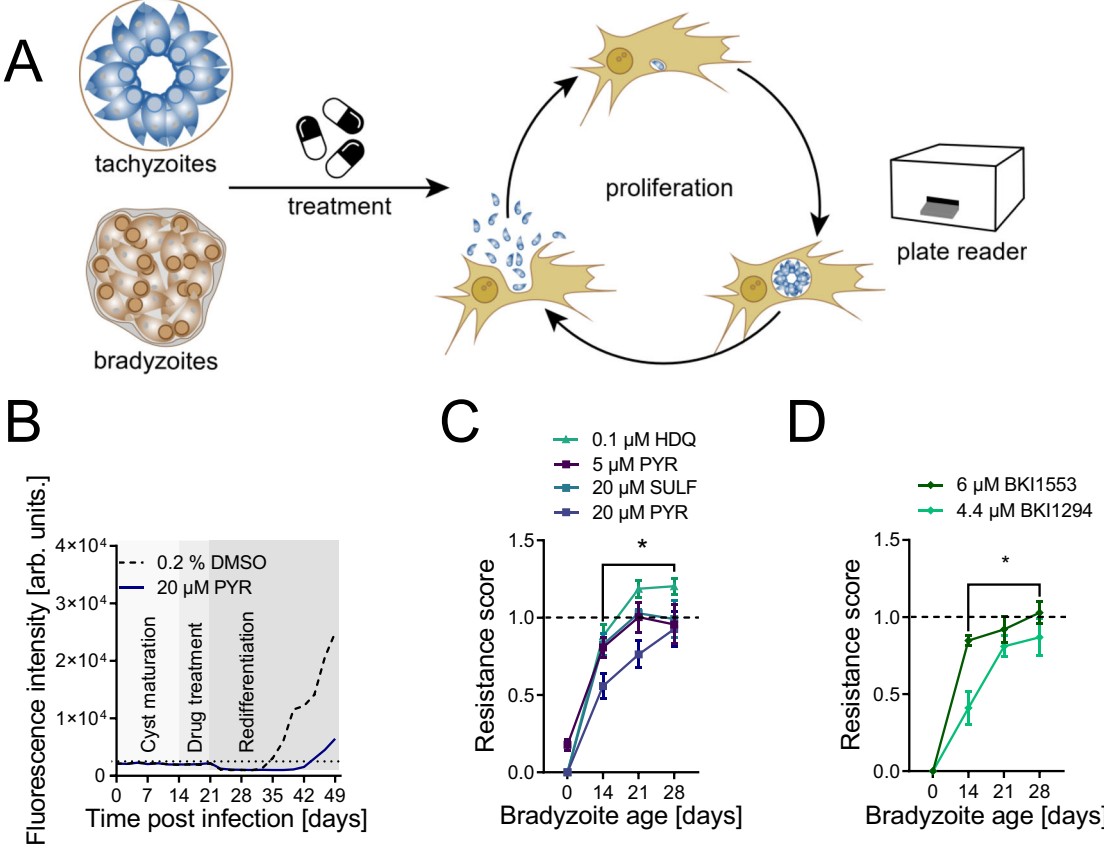

**Fig. 5 *T. gondii* in vitro bradyzoites develop broad drug tolerance. A** Experimental design for in vitro drug tolerance assay. Pru-tdTomato cysts and tachyzoite controls were drug treated at the indicated concentrations for one week. Following the treatment, medium was changed to bicarbonate-replete medium to induce re-differentiation to proliferating tachyzoites and tdTomato fluorescence was monitored. **B** Raw fluorescence intensities of a representative experiment during cyst maturation for 14 days, 7 days of treatment with 20 μM pyrimethamine or 0.2% DMSO as solvent control and re-differentiation for 28 days. Black dotted line indicates the limit of detection. Data represents the means of one experiment with five replicates. **C**, **D** Resistance scores of tachyzoites (0) and 14-, 21-, and 28-day-old bradyzoites treated with pyrimethamine (PYR), sulfadiazine (SULF) or 1-hydroxy-2-dodecyl-4(1*H*) quinolone (HDQ) and bumped kinase inhibitor (BKI) 1553 and BKI1294. Shown are means of three independent experiments with five replicates each and SEM (*$p = 0.0023$ for 20 μM pyrimethamine treatments, $p = 0.025$ for sulfadiazine treatment and $p = 0.0002$ for HDQ treatment, $p = 0.035$ for BKI1553 and $p = 0.0017$ for BKI1294 treatments, two-tailed Mann–Whitney *U* tests). Source data are provided as a Source Data file.

high doses of pyrimethamine increased until 28 days of maturation (Fig. 5C). In contrast, tachyzoite-infected myotubes, which served as a control and are shown as '0 days differentiated cultures', only marginally survived treatment with 5 μM pyrimethamine, with a low RS of 0.18. While resistance of bradyzoites to antifolates is well-established resistance to mitochondrial inhibitors is less well documented. We sought to test whether exclusion of HDQ by the cyst wall would lead to insensitivity and thus monitored resistance and mitochondrial membrane potential of 28-day-old cysts. Up to one 1 μM of HDQ did not decrease the RS and cyst diameter but decreased the mitochondrial membrane potential estimated by the drop of intensity of Mitotracker staining intensity by 60% (Fig. S7A–D). Furthermore, we tested the resistance against two different bumped kinase inhibitors (BKI), BKI1553 and BKI1294 (Fig. 5D), that have been shown to decrease cyst burden in chronically infected mice[45,46]. Consistent with previous results, tachyzoites did not grow after exposure to fourfold of the IC$_{50}$ of each compound. In contrast, already 14-day-old tissue cysts were resistant to both treatments, with an RS of 0.42 and 0.83, respectively. Resistance increased further and BKI1553 became completely ineffective against 28-day-old cysts while cysts reached an RS of 0.89 against BKI1294 (Fig. 5D).

Taken together, these data demonstrate that myotube-cultured cysts gradually develop resistance against both established and experimental antiparasitics and are suited for compound screens against chronically infectious forms of *T. gondii*.

**The metabolome of *T. gondii* depends on the host cell type and parasite stage.** The metabolome is directly linked to the mode of action of many antiparasitics and may also be associated with their efficacy[5]. However, the metabolome of bradyzoites and its differences to tachyzoites remains largely unknown. To test whether bradyzoites possess a distinct metabolome, we compared the abundance of metabolites between ME49 parasites in their bradyzoite form with intracellular tachyzoites in both HFF cells and myotubes (Fig. 6). Parasites were differentiated into cysts for 28 days, quenched with ice-cold phosphate-buffered saline (PBS), released from cells by needle passage, and purified from host cell debris via binding to DBA-coated magnetic beads. Intracellular tachyzoites were prepared by needle passage and filtered to remove host material. We also estimated artefacts that are introduced by differences in quenching and preparation procedures of tachyzoites and bradyzoites. We supplemented isolated tachyzoites with beads and processed them similarly to bradyzoites. To correct for background metabolites from contaminating host material and magnetic bead preparations, we included uninfected myotubes and bead-only controls into our

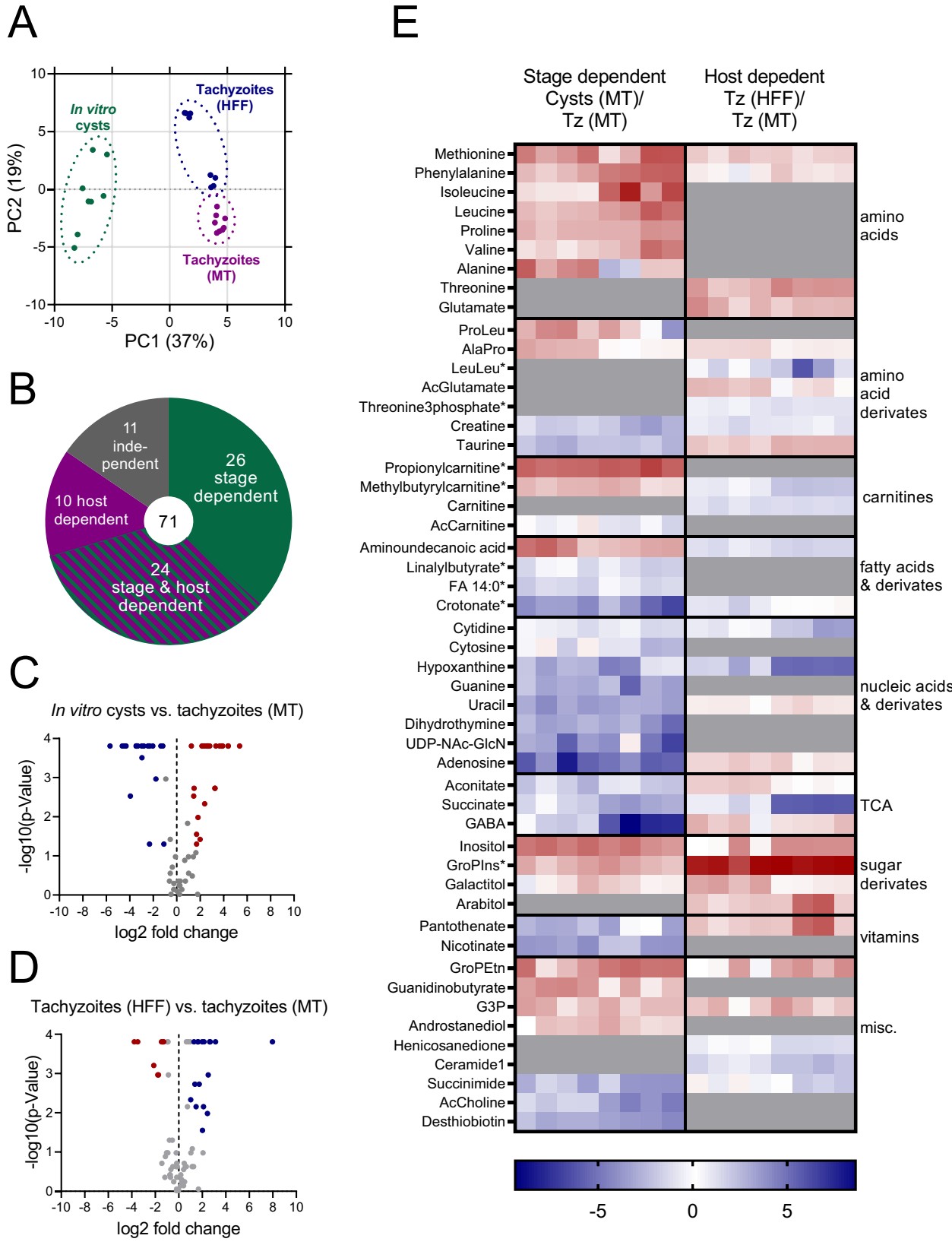

analysis. These samples were treated identically to bradyzoite samples. All preparations were extracted in 80% acetonitrile and analyzed on a HILIC-UHPLC-LC/MS. Blank and host cell background-subtracted ion intensities were normalized to the total ion count per sample and log2-fold changes of metabolites between experimental conditions calculated (Fig. S8A–F,

Supplementary Data 1). Principal component (PC) analysis indicated that bradyzoites in the purified tissue cysts differed from tachyzoites along PC1, while myotube- and HFF-derived tachyzoites were separated in PC2 (Fig. 6A). Bead-supplemented tachyzoite controls that underwent incubation with similar purification procedure as tissue cysts and were incubated with

**Fig. 6 _T. gondii_ in vitro cysts harbor a distinct metabolome. A** Principal component analysis of untargeted metabolomic data comparing 28-day-old ME49 _T. gondii_ in vitro cysts cultivated in KD3 myotubes, tachyzoites in KD3 myotubes (MT) and tachyzoites in human fibroblasts (HFF). Two individual experiments with four replicates each were analyzed. **B** Pie chart showing the number of metabolites within the dataset and how they were affected by host cell environment and parasite stage. **C, D** Volcano plots of log2-fold changes calculated from fractional metabolite abundances between **C** in vitro cysts and myotube-isolated tachyzoites or **D** tachyzoites isolated from HFFs and myotubes (n = 8, two-tailed Mann–Whitney U test). **E** Heatmap showing log2-fold changes of significantly different metabolites (_p_ < 0.05, uncorrected Mann–Whitney U test) between in vitro cysts and tachyzoites (Tz) in MTs (left panel) and tachyzoites in HFFs compared to tachyzoites in myotubes (right panel). Log2-fold changes were calculated pair-wise. Gray areas indicate non-significantly changed metabolites. Asterisks denote putative metabolite identifications based on accurate mass only. Ac acetyl, UDP-NAc-GlcN UDP-N-acetyl-glucosamine, GABA γ-aminobutyric acid, GroPIns glycerphosphoinositol, GroPEtn glycerophosphoethanolamine, G3P glycero-3-phosphate. Source data are provided as a Source Data file.

beads for one hour were indistinguishable from pure tachyzoites. Uninfected host cell controls were similar to bead-only samples, indicating minimal metabolite alterations and contamination due to the purification procedures (Fig. S8E). The two clusters of HFF-derived tachyzoites represent two independent experiments whose variation manifests itself after host-background subtraction (Fig. S8D). We quantified the levels of 71 metabolites, excluding those that we detected in uninfected controls. The levels of 26 metabolites varied between parasite stages, 10 varied between types of host cells, while 24 depended on both factors, and 11 remained invariant (Fig. 6B–D). Comparing bradyzoites to tachyzoites from the same host cell type (Fig. 6E), we noted that the relative abundances of amino acids and sugar derivatives were generally increased, while vitamins, derivatives of amino acids, nucleic acids and metabolites associated with the TCA cycle were significantly decreased. Fatty acids (FAs) and related molecules are variably affected by stage conversion. When comparing tachyzoites from HFF cells with those from myotubes (Fig. 6E), we found that the levels of amino acids and sugars were higher in HFF-grown tachyzoites, while FAs increased in abundance. The host cell type had mixed effects on other metabolite classes.

These data reveal distinct metabolic features in _T. gondii_ bradyzoites involving various metabolite classes and the mitochondrial TCA cycle.

**Mature _T. gondii_ tissue cysts develop resistance against the aconitase inhibitor sodium fluoroacetate.** We hypothesized that diminished pools of TCA metabolites in bradyzoites would be consistent with lower importance in this stage. To estimate the reliance of bradyzoites on the TCA cycle, we used the aconitase inhibitor NaFAc to inhibit the TCA cycle flux in _T. gondii_[6]. Continuous exposure to NaFAc arrests tachyzoite replication with an $IC_{50}$ of 168 μM, and parasite replication was minimal at 500 μM (Fig. 7A). Next, we tested the ability of tachyzoites and bradyzoites to survive exposure to 0.5–5.0 mM NaFAc for seven days. 28-day-old cysts recovered after treatment with a RS of approximately 0.8, while tachyzoites were only able to survive 0.5 mM NaFAc, but not higher concentrations (Fig. 7B). To exclude that tolerance of bradyzoites toward NaFAc relies on exclusion of this inhibitor from the cyst[47], we sought to measure marked direct metabolic effects of this inhibitor on tachyzoite and bradyzoite stages[6]. We treated 28-day-old ME49 cysts and intracellular myotube-cultured tachyzoites with NaFAc for seven and three days, respectively, and analyzed the metabolomes by HILIC-UHPLC-LC/MS (Fig. 7C, D, Supplementary Data 2). The PC analysis revealed that solvent- and NaFAc-treated tachyzoites form distinct clusters that are separated along the PC2 axis. In contrast, treatment of tissue cysts did not lead to a separation from untreated controls (Fig. 7C). NaFAc treatment affected the relative abundance of 33 metabolites in tachyzoites, but changed only 8 metabolites in bradyzoites (Fig. 7D). Aconitate levels in both parasite stages were increased significantly upon NaFAc

treatment (Fig. 7D). These data are consistent with a NaFAc-mediated inhibition of aconitase in bradyzoites and show that extended exposure to high doses of this inhibitor, in contrast to tachyzoites, is well tolerated.

## Discussion

Persistence mechanisms of important protozoan pathogens, including _T. gondii_, _T. cruzi_, _Leishmania_ spp. and _P. vivax_, remain largely understudied, and there are inadequate or no treatments available against chronic infections with these organisms[2]. The limited availability of methods to culture persister cells restricts the approaches to study the link between persistence and metabolism[2,48]. Here, we report a cell culture model based on immortalized human skeletal muscle cells[36] that supports maturation of tissue cysts of type I, II and III _T. gondii_ strains. These myotube-grown bradyzoites exhibit key characteristics including typical ultrastructural features, stress resistance, and tolerance toward antimicrobials.

Our in vitro system is scalable, enabling measurements of bradyzoite metabolites via mass spectrometry. To this end, we established a protocol that allows isolation and purification of cysts in large quantities and without the use of Percoll gradients[22], making them available for mass spectrometry-based metabolomics. We obtained an estimated yield of $2 \times 10^6$ cysts per two T150 culture dishes that were sufficient for one technical replicate and represented a volume-adjusted tenfold improvement to the bradyzoite yield from the infection of one mouse[22]. PC analysis of the impact of host cell type and the parasite stage indicates mostly independent effects on the metabolome (Fig. 5A). Stage conversion leads to changes that are largely homogeneous across classes of metabolites such as amino acids, vitamins, TCA cycle intermediates, and nucleic acids. This indicates the operation of distinct metabolic homeostasis mechanisms in bradyzoites.

Bradyzoites harbor lower levels of several TCA cycle-associated metabolites that are consistent with a lower reliance on this pathway (Fig. 6E). This interpretation is supported by an NaFAc-induced slight accumulation of aconitate (Fig. 7B). Extensive accumulation of citric acid and aconitate has previously been shown to occur in extracellular tachyzoites already after four hours of treatment[6]. We attribute the lower accumulation of aconitate we observe in bradyzoites to the extended treatment over seven days and the longer quenching and purification procedure required for intracellular parasites. Importantly, this accumulation of aconitate suggests that NaFAc is available to bradyzoites, that the aconitase is active and inhibited by NaFAc. This interpretation also agrees with previous findings of an active TCA cycle enzyme, isocitrate dehydrogenase, in lysates of four-week-old bradyzoites from mice[49] and with continued transcription of TCA-cycle-related genes[50]. Interestingly, NaFAc exerts comparatively limited effects on other non-TCA cycle-related metabolites in bradyzoites. It is noteworthy that our data do not exclude the possibility that parts of the TCA cycle remain

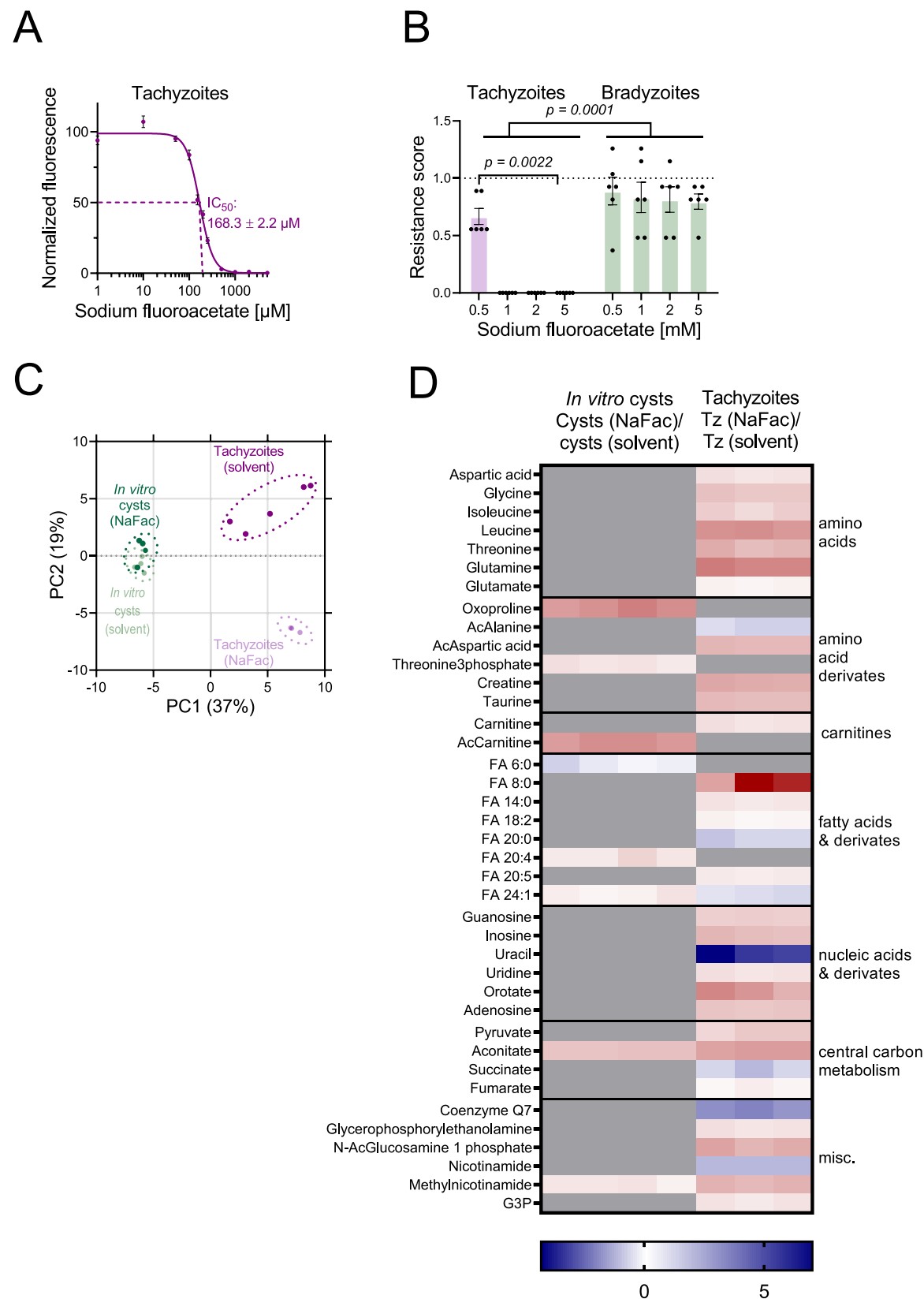

operational and may process anaplerotic substrates, such as glutamine to generate NADH for subsequent ATP production through the mETC. However, full resistance to the inhibitor of the alternative type 2 NADH-dehydrogenase TgNDH2[51] (Figs. 5C and S7C, D) suggests that this source of ATP would equally be dispensable. HDQ appears to only partially decrease

the mitochondrial membrane potential in bradyzoites (Fig. S7A, B), indicating the presence of HDQ-independent sources of reduction equivalents or a very slow rate of mitochondrial ATP synthesis concomitant with negligible depolarization of the membrane potential. Bradyzoites may rely increasingly on glycolytic ATP production, as indicated by increased activity of

**Fig. 7 Bradyzoites develop resistance to an aconitase inhibitor. A** Calculation of half inhibitory concentration (IC$_{50}$) of sodium fluoroacetate (NaFAc) for Pru-tdTomato tachyzoites. Parasites were grown for 7 days in presence of NaFAc and the IC$_{50}$ value was calculated from fluorescence intensities normalized to untreated controls. Shown are means and SEM of three independent experiments with three replicates each. **B** Resistance scores of tachyzoites and 28-day-old bradyzoites treated with NaFAc or 0.5% water as solvent control. Shown are means and SEM of two and three independent experiments for bradyzoites and tachyzoites performed in triplicates, respectively, two-tailed Mann–Whitney *U* test, *p* < 0.0001. **C** Principal component analysis of untargeted metabolomic data comparing 28-day-old ME49 *T. gondii* in vitro cysts cultivated in KD3 myotubes either incubated for 7 days in presence of 1 mM NaFAc or an additional 0.5% water as solvent control and tachyzoites cultivated in myotubes treated for three days with 500 µM NaFAc or 0.25% water as solvent control. One experiment with four replicates for bradyzoite groups, five replicates for solvent treated, and three replicates for NaFAc-treated tachyzoites were analyzed. **D** Heatmap showing log2-fold changes of significantly different metabolites (*p* < 0.05, uncorrected Mann–Whitney *U* test) between NaFAc-treated and vehicle-treated in vitro cysts and between NaFAc-treated (left panel) and vehicle tachyzoites (Tz) (right panel). Log2-fold changes were calculated pair-wise. Gray areas indicate non-significantly changed metabolites. Ac acetyl, G3P glycerol 3-phosphate. Source data are provided as a Source Data file.

glycolytic enzymes in this stage[49]. The number of *T. gondii* cysts in brain tissue of latently infected mice depends on the turnover of large amylopectin pools[52] and on the expression of several glycolytic enzymes including hexokinase[11] and lactate dehydrogenase[12,53]. Also, the pyruvate dehydrogenase E1 subunit (*pdhA*) transcript is less abundant in four-week-old bradyzoites[54]. Interestingly, transcriptomic analyses of *P. vivax* hypnozoites, a persistent stage of this malaria parasite, indicates a similar metabolic shift toward glycolysis[55].

We also found that a number of amino acids were accumulated in bradyzoites (Fig. 6E). We attribute these changes directly to stage conversion and not to differences in media composition. The affected amino acids, except for proline, which can also be synthesized from glutamate[56], are less concentrated in the Roswell Park Memorial Institute (RPMI)-based bradyzoite medium compared to the DMEM-based medium used for tachyzoites. All of these amino acids have been shown to be imported by extracellular tachyzoites[57]. *T. gondii* tachyzoites express several Apicomplexan Amino Acid Transporter proteins[57], all of which remain transcriptionally expressed in bradyzoites[54] (Fig. S9). This suggests that bradyzoites retain the ability to take up amino acids from their environment. Interestingly, restriction of some exogenous amino acids has been used to facilitate stage conversion[48,58] suggesting that alternative sources may represent adaptions to limiting supply of exogenous amino acids. These include obligate autophagy and proteolysis through autophagy-related protein (ATG9)[19] and cathepsin L[59] and lower amino acid demand by translational repression through the eukaryotic initiation factor 2α[60]. The diversification of nutrient sources would resemble the stringent metabolic response of *Leishmania mexicana* amastigotes that also adopt a slow-growing phenotype within the phagolysosome of macrophages[61].

Nucleobase-containing metabolites were found to be less abundant in bradyzoites (Fig. 6E). While the parasite is auxotroph for purines, pyrimidines can both be imported and synthesized[62]. Slowly growing or dormant bradyzoites likely require lower amounts of nucleobases. Consistently, the transcription of the corresponding low affinity and high-capacity adenosine transporter (*TgAT*)[63,64] appears to be down-regulated in 4-week-old bradyzoites in vivo (Fig. S9). Two other proteins containing nucleoside transport signatures remain expressed and may represent the unidentified high-affinity purine transporter being responsible for maintaining a slow influx of nucleobases[54] (Fig. S9). Pyrimidine salvage does not appear to be essential for bradyzoite formation[65] indicated by its continued synthesis. Similar to *P. falciparum*[66], pyrimidine synthesis in *T. gondii* tachyzoites depends on the mETC to provide an electron acceptor to the DHOD[67]. The role and relevance of the mETC in bradyzoites remains to be formally established. While we show that in vitro bradyzoites develop tolerance toward the dual alternative NADH dehydrogenase and DHOD inhibitor HDQ[43,68]

(Fig. S7D), other mETC inhibitors, such as atovaquone appear to have some effect and lead to a decrease in cyst counts in brains of latently infected mice[15–17,69]. Together, available data suggest that the mETC is a viable target in tissue cysts but that the importance of the associated metabolic pathways, such as the TCA cycle may differ from tachyzoites.

Multinucleated myotubes are generally arrested in the G0 phase of the cell cycle and exhibit highly branched mitochondria and oxidative metabolism as opposed to glycolytic lactate production[70]. Both traits have been shown to also facilitate stage conversion of *T. gondii*[27,71,72] and may contribute to the suitability of KD3 cells to harbor *T. gondii* tissue cysts by limiting the expansion of non-converting tachyzoites. While the differentiation efficiency of all tested *T. gondii* strains can be further increased by additional non-physiological stressors such as basic pH, it is not needed for extended cultures. Strikingly, even highly virulent RH-derived parasite sub-strains did not overgrow the culture (Fig. 2C). Resistance of bradyzoites against pepsin digestion and temperature stress develops after 14 days post infection and increases until 28 days post infection (Fig. 3C). The onset of tolerance toward inhibitors of well-established drug targets including the calcium-dependent protein kinase 1 (TgCDPK1)[45,46], dihydrofolate reductase-thymidylate synthase (TgDHFR-TS)[73] and the DHOD (Fig. 5C, D) follows similar timing and is shared with *T. gondii* cysts in the brain of chronically infected mice[44–46,74]. Hence, our in vitro model will facilitate dissecting the molecular basis of the impressive drug tolerance and stress resistance of *T. gondii* tissue cysts. Expectedly, in vitro cysts are also orally infectious to mice (Fig. 4A-D). Comparing estimated volumes of in vitro cysts and in vivo brain-derived cysts, our data suggest that 10 in vivo cysts are equivalent to 500 in vitro cysts in this regard. Surprisingly, 3 out of 20 mice that received tachyzoites via oral gavage seroconverted. It has been observed earlier that numbers greater than 1000 pepsin-sensitive tachyzoites were orally infectious to mice[75].

Electron microscopic analysis of the ultrastructure of myotube-derived tissue cysts revealed typical features such as a developing cyst wall, amylopectin granules, and posteriorly positioned nuclei in the bradyzoites (Figs. 2A, S1)[76]. Interestingly, we also found evidence of disintegrating bradyzoites at all time points up to 21 days post infection, (Fig. S1) which has also been described previously in mice after four and 8 weeks of infection[22,76,77]. This parasite turnover likely contributes to the heterogeneous appearance with respect to cyst size and bradyzoite packaging density[21,22]. A similar phenotypic heterogeneity has been ascribed to the ability of other intracellular pathogens to persist in the presence of stressors such as drugs and immune effectors[78].

In conclusion, we present a novel system to culture functionally mature *T. gondii* tissue cysts that surpasses typical in vivo systems through its scalability. Importantly, it allows cyst purification with methods compatible with subsequent mass spectrometry-based

applications. We show that bradyzoites maintain a distinct metabolome that confers resistance to an aconitase inhibitor. This culture model will be valuable for uncovering new bradyzoite biology and for addressing drug tolerance.

## Methods

**Host cell lines and cultivation**. All cultures were maintained in a 37 °C humidified $CO_2$ (10%) incubator as described previously[7].

Briefly, BJ-5ta human foreskin fibroblast (ATCC CRL-4001 HFF) monolayers were cultured in Dulbecco's Modified Eagle's Medium (DMEM) (Gibco) supplemented with 25 mM glucose (Sigma-Aldrich), 4 mM L-glutamine (Thermo Fisher Scientific), 1 mM sodium pyruvate (Capricorn Scientific), 100 U/ml penicillin, 100 µg/ml streptomycin (Thermo Fisher Scientific) and 10% heat-inactivated bovine serum (Capricorn Scientific).

The immortalized human myoblast cell line KD3 (a kind gift from N. Hashimoto; Department of Regenerative Medicine, National Institute for Longevity Sciences, National Center for Geriatrics and Gerontology, Oobu, Japan[36]), initially derived from normal subcutaneous female muscle tissue, was cultured in DMEM supplemented with 25 mM glucose, 4 mM L-glutamine, 1 mM sodium pyruvate, 100 U/ml penicillin, 100 µg/ml streptomycin, 2% Ultroser G (Cytogen GmbH) and 20% heat-inactivated fetal bovine serum (FBS) (Capricorn Scientific). The differentiation of myoblasts into myotubes was induced at 70% confluency by changing medium to DMEM supplemented with 25 mM glucose, 4 mM L-glutamine, 1 mM sodium pyruvate, 100 U/ml penicillin, 100 µg/ml streptomycin, 2% horse serum (HOS) (Capricorn Scientific), 10 µg/ml human insulin (Sigma-Aldrich), 5 µg/ml human holo-transferrin (PAN Biotech) and 1.7 ng/µl sodium selenite (Sigma-Aldrich) for 5–7 days[36].

**Parasite strains cultivation and differentiation**. Type I strain RH[79], RH-Δku80Δhxgprt (RHΔku80)[80], GT1[81], Type II ME49[82], NTE[82], Pru-Δku80Δhxgprt(BSG-4) (Pru-GFP)[42], Pru-Δhxgprt tdTomato (Pru-tdTomato)[83] and Type III NED[84], VEG ([85], provided by G. Schares), were maintained in vitro in HFF monolayers grown in DMEM supplemented with 25 mM glucose, 4 mM L-glutamine, 1 mM sodium pyruvate, 100 U/ml penicillin, 100 µg/ml streptomycin and 1% heat-inactivated FBS (tachyzoite medium). Freshly egressed parasites were passaged by transfer to new HFF monolayers.

Differentiation of tachyzoites into tissue cysts was facilitated by $CO_2$ depletion at pH 7.4 if not otherwise indicated. To avoid acidification, the medium was changed to low glucose (5 mM), 50 mM HEPES (Sigma-Aldrich) buffered RPMI 1640 medium (Gibco) supplemented with 4 mM L-glutamine, 100 U/ml penicillin, 100 µg/ml streptomycin, 2% HOS, and 10 µg/ml human insulin, 5 µg/ml human holo-transferrin and 1.7 ng/µl sodium selenite (bradyzoite medium). The cells were incubated at 37 °C and ambient $CO_2$ levels. Medium was changed every two days and cells were washed once per week with PBS. The day after infection, infected monolayers were washed with prewarmed PBS to remove non-invaded parasites. Medium was changed every second day and bradyzoite cultures were washed with PBS once a week. For assays involving plate reader-based fluorescence measurements, phenol red was omitted from the medium.

**Electron microscopy**. KD3 myotube cultures in T60 dishes were infected with 1.3 × 10⁶ Pru-tdTomato tachyzoites corresponding to a multiplicity of infection (MOI) of 0.3 and bradyzoite formation was facilitated for the indicated times in bradyzoite medium at ambient $CO_2$ levels. On the day of the experiment, the medium was removed and samples were fixed by covering the monolayer with 1% paraformaldehyde (Sigma-Aldrich) and 2.5% glutaraldehyde (Sigma-Aldrich) in 0.05 M HEPES buffer (pH 7.4). After incubation at RT for 3 h, samples were sealed with parafilm and stored in the fridge until further processing.

For plastic embedding, cells were scraped off with a cell scraper, sedimented by centrifugation (3000 × g, 10 min), and washed twice with 0.05 M HEPES buffer for removing the fixative. The washed cell pellets were mixed with 3% low-melting point agarose (1:1 [v/v]) at 40 °C, centrifuged (3000 × g, 5 min) and cooled on ice. The cell pellets were cut-off from the agarose gel block using a sharp razor blade and stored in 2.5% glutaraldehyde in 0.05 M HEPES buffer. Post-fixation, block contrasting and dehydration, embedding in epoxy resin was performed according to a standard procedure[86]. Ultrathin sections with a thickness of ~65 nm were generated with an ultramicrotome (UC7, Leica Microsystems, Germany) using a diamond knife (45°, Diatome), collected on copper slot grids and contrasted with 2% uranyl acetate (Thermo Fisher Scientific) and 0.1% lead citrate (Sigma-Aldrich).

Electron microscopy of ultrathin sections was performed with a transmission electron microscope (Tecnai Spirit, Thermo Fisher Scientific) at 120 kV. Images were recorded with a side-mounted CCD camera (Megaview III, EMSIS, Germany) and the montage function of the camera software (Multiple Image Alignment, iTEM 5.2 (Build 3468), EMSIS, Germany) was used to cover larger field of views with sufficient pixel resolution.

**Immunofluorescence assay**. Uninfected host cells were grown in monolayers on 12 mm round glass coverslips (TPP) and infected for 24 h, unless indicated otherwise. For bradyzoite experiments, independent of strain and host cell

background, myotube monolayers were infected with 1.3 × 10⁴ extracellular tachyzoites corresponding to an MOI of 0.1 and incubated for the indicated times. Samples were fixed with 4% paraformaldehyde in PBS for 20 min at RT followed by a washing step in 0.1 M glycine/PBS (Thermo Fisher Scientific) and permeabilization for 20 min in 0.1% Triton X-100 (Thermo Fisher Scientific) in PBS. Cells were blocked for 1 h in 2% bovine serum albumin (BSA) (Thermo Fisher Scientific) in PBS, stained for 1.5 h with primary and for 1 h with species-matched secondary antibodies at RT, and mounted on microscopy slides (Thermo Fisher Scientific) in Fluoromount-G (Sigma-Aldrich) containing DAPI (1:3,000) (Thermo Fisher Scientific). Samples were first stained with biotinylated DBA 1:1,000 (Sigma-Aldrich) and primary monoclonal antibodies (mab) anti-rat CC2 1:1,000[40]; anti-mouse myosin heavy chain (MF20) 1:400 (eBioscience™); anti-mouse SAG1 mab DG52 1:500[87] and then with goat anti-rat-Alexa546 1:300 (Invitrogen), goat anti-rat-Alexa350 1:300 (Thermo Fisher Scientific), rabbit anti-mouse-Cy3 1:400 (Dianova), donkey anti-mouse-Cy5 1:300 (Dianova) or streptavidin-Alexa488 1:2,500 (Dianova), and streptavidin-Cy5 1:2,500 (Jackson ImmunoRearch). Mitochondrial staining was done prior to fixation for 30 min with Mitotracker™ Deep Red FM (Thermo Fisher Scientific) at 200 nM in tachyzoite medium at 37 °C in a humidified $CO_2$ incubator. Cultures were chased for 15 min in tachyzoite medium and washed three times with PBS, fixed with 4% paraformaldehyde followed by immunofluorescence staining as indicated. Monochromatic images were recorded with a Zeiss Apotome Imager equipped with a Plan-Fluar 63x/1.45 oil M27 objective or on a Zeiss Axio Observer imager equipped with Plan-NeoFluar 10x/0.3 objective or with a Plan-Apochromat" 63x/1.4 oil objective and imported into ImageJ version 1.52a[88] for coloring, the generation of overlays and quantification.

**Pepsin digestion and in vitro oral transmission assay**. KD3 myotube cultures in 6-well plates were infected with 6 × 10⁴ extracellular Pru-tdTomato tachyzoites corresponding to an MOI of 0.1 and bradyzoite formation was induced for the indicated times in bradyzoite medium. Tachyzoite controls were infected for two days with 3.2 × 10⁶ parasites corresponding to an MOI of 5 in tachyzoite medium. On the day of experiment, pepsin digestion was performed for 20, 40, and 60 min at 37 °C as described previously[75] using a modified pepsin solution. The buffered solution consisted of 1.01 g pepsin (Thermo Fisher Scientific), 0.166 g glycine (Thermo Fisher Scientific), 0.129 g NaCl (Sigma-Aldrich) and 17.8 ml of a 1 M HCl solution (Sigma-Aldrich), filled up to 200 ml with Milli-Q-water, resulting in a pH of 1.2 and supplemented with 5 mM glucose. Digestion was terminated by neutralization using a 1.2% sodium bicarbonate solution (pH ≈ 8.3) containing 0.0159 g/l phenol red as pH indicator dye. Digested infected monolayers and uninfected and undigested controls were collected with a cell scraper and harvested via centrifugation (1200 × g, 10 min). The supernatant was then removed, the pellets were resuspended in 450 µl DMEM tachyzoite medium and used to infect HFF cells in a 96-well plate in triplicates, if not indicated differently. Parasite recovery was monitored by measuring tdTomato fluorescence intensity (excitation 554 nm/emission 589 nm) in a plate reader (Tecan Infinite M200 PRO) using Tecan i-control software v2.0.10.0 every second day for up to 21 days. Medium was carefully exchanged without perturbating infected monolayers after each measurement.

For the oral transmission assay, scraped monolayers and controls were challenged by the indicated combination of stresses: cold stress (4 °C for 4 and 7 days), heat stress (55 or 65 °C for 5 min), incubation at RT for two days, and pepsin digestion (60 min).

The RS was calculated as the difference of days passed between the detection of vehicle-treated and treated tachyzoite cultures. This difference was normalized to the number of days of total observation to account for differences in initial cyst loads and the resulting re-growth periods between experiments:

$$RS = 1 - (\#days_{treatment} - \#days_{vehicle})/\#days_{observed}$$

An RS of 0 indicates no recovery from the treatment and a RS of 1 means equal growth to the vehicle-treated control, indicating full resistance to the treatment.

**In vivo oral transmission assay**. KD3 myotube cultures in 6-well plates were infected with 1.9 × 10⁵ Pru-tdTomato tachyzoites corresponding to a MOI of 0.3. Bradyzoite formation was induced for indicated times in bradyzoite medium. Tachyzoite cultures were prepared in T25 cell culture flasks by infecting KD3 myotubes and human fibroblasts with 3.6 × 10⁶ tachyzoites corresponding to an MOI of 2 for 2 days in tachyzoite medium. The cultures were then prepared for shipping and sent to Freiburg at RT within 24 h. The cultures were washed once with PBS, scraped in fresh respective medium, and harvested via centrifugation (1200 × g, 10 min). The supernatant was removed after centrifugation and the pellet was resuspended in 2 ml PBS and placed on ice. For tachyzoite controls, scraped monolayers were syringed with a 27 G needle (Sterican®) to release intracellular parasites and counted in a Neubauer chamber (Roth). For cyst quantification, 10 µl of the cyst solution was placed on a glass slide and cysts were counted based on the fluorescence of tdTomato in a fluorescence microscope. For in vivo cyst generation, one C57BL/6JRj mouse was infected with 1 × 10³ T. gondii tachyzoites via intraperitoneal injection. Thirty days post infection, the mouse was euthanized by cervical dislocation. The brain was harvested in 2 ml PBS and minced using an 18G and 21G needle (Sterican®) for cysts inspection via DBA staining as described previously[89]. Briefly, 50 µl out of 200 µl DBA-stained sample was pipetted in a

clear-bottomed 96-well plate and quantified at ×20 magnification using DBA positivity as a first criterion and tdTomato positivity to confirm brain cyst.

**Mouse infections**. Groups of five 10-week-old male and female mice (C57BL/6JRj) were obtained from certified breeders (Janvier Labs, Route du Genest, 53940 Le Genest-Saint-Isle, France) and kept under specific-pathogen-free conditions in the local animal facility (Department for Microbiology and Hygiene, Freiburg). Mice were kept on a day/night cycle of 14 and 10 h at 20–22 °C and a relative humidity between 45% and 55%. Mice were fed keeping diet (Altromin) and non-autoclaved drinking water ad libitum and housed in green line type 2 cages (Tecniplast). All animal experiments were performed in accordance with the guidelines of the German animal protection law and the Federation for Laboratory Animal Science Associations. Experiments were approved by the state of Baden-Württemberg (Regierungspräsidium Freiburg; reference number 35-9185.81/G-19/89).

Mice were infected by oral gavage with 50 or 500 freshly prepared in vitro-generated Pru-tdTomato cysts, 10 freshly prepared in vivo-derived Pru-tdTomato cysts or $10^4$ myotube- or HFF-derived extracellular Pru-tdTomato tachyzoites in a total volume of 200 μl sterile PBS.

Infected mice were monitored and weighed daily for the duration of the experiment. Relative weight loss was calculated based on the weight at the day of infection. Brains and blood from moribund animals were collected starting from 12 days post infection. After 30 days post infection, blood samples of all surviving mice were collected from the facial vein. Blood was allowed to clot at RT for 30 min before centrifugation (500 × g, 10 min). Serum was collected and either immediately analyzed or stored at −20 °C until further analysis. Animals were euthanized by cervical dislocation. Brains were harvested and cysts per brain sample were blindly counted as described previously:

$$\#Cysts/brain = \text{cysts in } 50\,\mu l \times 40$$

Seroconversion was evaluated via ELISA as described previously[89]. Serum from non-infected (n.i.) mice were included to calculate the cut-off:

$$cut\text{-}off = mean\,Absorbance^{450nm-570nm}(n.i.)$$
$$+ 2 \times standard\,deviation\,of\,Absorbance^{450nm-570nm}(n.i.)$$

An individual cut-off was calculated per each individual ELISA. In order to compare the *T. gondii* specific-antibody titers from different independent experiments a reactivity index was calculated per sample:

$$Reactivity\,index = mean\,Absorbance^{450nm-570nm}/cut\text{-}off$$

Reactivity index values higher than one were considered as positive.

**Cyst isolation**. Uninfected and infected myotubes were prepared. In both cases, two T150 dishes were pooled into one sample. For bradyzoite samples, myotubes were infected with $3.2 \times 10^6$ Pru-tdTomato tachyzoites corresponding to a MOI of 0.3 and cyst formation was induced for indicated time. On the day of harvest, infected samples and uninfected host cell controls were placed on ice, medium was removed and monolayers were washed three times with ice-cold PBS. Cells were then harvested by scraping into 10 ml ice-cold 0.05% BSA in PBS per T150 dish. Cysts were released from the monolayer via forcing through a 23 G needle (Sterican®) 25 times with a syringe and collected via centrifugation (1200 × g, 10 min, 0 °C). The supernatant was removed, the pellet was resuspended carefully in ice-cold 2% BSA in PBS containing 200 μl DBA-coupled beads (preparation described below) and samples were incubated for 1 h at 4 °C with gentle shaking. Subsequently, the samples were placed in a magnetic stand on ice, washed five times with 0.1% BSA in PBS to remove cell debris, followed by two washing steps with PBS to remove residual BSA. Cysts and beads were then collected via centrifugation (1200 × g, 10 min, 0 °C), shock frozen in liquid nitrogen and stored at −80 °C until extraction. Tachyzoite samples were generated in T150 dishes by infecting myotubes and HFF cells with $3.2 \times 10^7$ tachyzoites corresponding to an MOI of 3 for 48 h. Medium was replaced by ice-cold PBS and monolayers were scraped and passaged through a 27 G needle. Tachyzoites were filter-purified through a 3 μm filter (Whatman) and PBS-washed by centrifugation (1200 × g, 10 min, 0 °C) three times. All samples were extracted simultaneously in 80% acetonitrile for LC/MS analysis as described below. Bead-only controls were processed equally. Bead-supplemented tachyzoite controls were processed equally to cyst samples, replacing washing steps via magnetic stand by centrifugation (1200 × g, 10 min, 0 °C).

**Preparation of beads**. Coupling of Dynabeads™ MyONE™ Streptavidin T1 (Thermo Fisher Scientific) to DBA was done as described in the manufacturer's protocol. Briefly, 200 μl beads/sample were resuspended in 1 ml PBS by vortexing, washed three times with PBS in a magnetic stand, and resuspended in 1 ml PBS containing 50 μg DBA/sample. The tube containing the DBA-magnetic bead mixture was incubated on a rotary mixer for 45 min at RT. Uncoupled DBA was removed by washing the coated beads three times with PBS. After washing, the DBA-coated beads were resuspended in 2 ml PBS containing 2% BSA.

**Untargeted metabolic analysis of in vitro cysts**. For metabolome measurements of tissue cysts and tachyzoites, tachyzoite isolation, cyst maturation and isolation were performed as described above. Metabolites were extracted in 80% acetonitrile (Carl Roth) and 20% water (Carl Roth) containing internal standards (phenolphthalein, CAPS, PIPES (Sigma-Aldrich)). Cell pellets were sonicated for 5 min and after centrifugation (21,500 × g, 5 min, 0 °C), the supernatants were transferred to MS vials for immediate LC/MS analysis. 5 μl of each sample were collected to generate a pooled biological quality control (PBQC). Twenty microliters of the in vitro cysts, bead control and host cell background samples, and 5 μl of the tachyzoite samples were injected. The injection order of the samples was randomized, blanks and PBQCs were injected periodically. The samples were analyzed on a Q-Exactive Plus mass spectrometer (Thermo Fisher Scientific) via 70k MS1 scans, with intermittent 35k data-dependent 35k MS2 scans in positive and negative mode separately.

Chromatographic separation was achieved on a Vanquish Flex fitted with an ACQUITY UPLC BEH Amide column (Waters). Running a 25 min linear gradient starting with 90% eluent A (10 mM ammonium carbonate in acetonitrile)/10% eluent B (10 mM ammonium carbonate in water) and ending with 40% eluent A/60% eluent B, followed by washing and equilibration steps.

XCalibur 4.2.47 software and Compound Discoverer 3.1 software (Thermo Fisher Scientific) was used for recording and for peak detection, combination of adducts, and compound annotation, respectively. Metabolite identifications were based on either retention time and accurate mass match to an in-house library of 160 authentic standards, or by matching accurate mass and MS2 fragments to *m/z* cloud database (mirrored offline in *m/z* vault v2.1.22.15 in May 2020) (Thermo Fisher Scientific). Data were exported to Excel for grouping, combination of datasets from positive and negative ionization runs and blank subtraction. Compounds with a coverage less than 50% in at least one sample group were excluded. All missing values were gap-filled with either the respective mean of each sample group or the global minimum intensity when undetected in a sample group. After normalization to the internal standard, the fractional abundances for each sample were calculated and centered between 1 and -1.

Raw data, data curation steps, and performed statistics are accessible in Supplementary Data 1 and Supplementary Data 2.

**Drug tolerance assays**. Myotubes in 96-well plates were infected with $2.4 \times 10^3$ *T. gondii* Pru-tdTomato tachyzoites corresponding to a MOI of 0.1 and cyst formation was induced for indicated times in bradyzoite medium at ambient $CO_2$ levels, followed by a drug pulse of 7 days with respective vehicle as solvent control. Tachyzoite controls were infected with $2.4 \times 10^4$ Pru-tdTomato tachyzoites corresponding to a MOI of 1 in tachyzoite medium for 2 h before the seven-day drug pulse was applied in tachyzoite medium. Parasite recovery was monitored by measuring tdTomato fluorescence for 28 days as described above. Medium was exchanged after each measurement. The RS was calculated as described above for pepsin digestion and the oral transmission assay.

(i) Developing drug tolerance. Cysts were generated for indicated times and treated with established antiparasitics (pyrimethamine, sulfadiazine (Sigma)) and HDQ (a kind gift of Wolfgang Bohne), two bumped kinase inhibitors (BKI1294, BKI1553)[90,91] or 0.2% DMSO (Thermo Fisher Scientific) as solvent control at the indicated concentrations for seven days. Respective tachyzoite controls were generated by infecting KD3 myotubes for 2 h with $2.4 \times 10^4$ tachyzoites corresponding to a MOI of 1 in tachyzoite medium before treatment.

(ii) Drug resistance of matured cysts vs. tachyzoites. Cyst formation was induced for indicated time and pulsed for seven days with indicated concentrations of sodium fluoroacetate (abcr GmbH) or 0.5% $H_2O$ as solvent control. Respective tachyzoite controls were generated as described above.

(iii) $IC_{50}$ assay tachyzoites. Growth of tachyzoite controls treated with the indicated concentrations of sodium fluoroacetate was monitored throughout the seven-day drug pulse. The half inhibitory concentration ($IC_{50}$) values were determined at day seven by fitting the dose-response curve by non-linear regression in GraphPad Prism v8 and v9.

**Statistical analysis**. All tests, except statistics of metabolite abundances, were performed in GraphPad Prism v8 and v9 as detailed in the results section. Statistical significance of metabolite changes was calculated in Microsoft Excel. PC analyses were computed using ClustVis 2.0[92].

**Reporting summary**. Further information on research design is available in the Nature Research Reporting Summary linked to this article.

## Data availability

Authors can confirm that all relevant data are included in the paper and/or its supplementary information files. The *m/z* cloud database is available here: https://www.mzcloud.org/. ToxoDB is available at https://toxodb.org/. The LCMS data generated in this study have been deposited in the metabolomics workbench database (https://www.metabolomicsworkbench.org) under accession code Study ID ST002051 and

ST002053 https://doi.org/10.21228/M82H7B; https://doi.org/10.21228/M82H7B. Source data are provided with this paper.

## Code availability

The code of CCEAN tool is available here: https://github.com/neumann-alexander/ccean; https://doi.org/10.5281/zenodo.552830.

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

## Acknowledgements

We are grateful to Naohiro Hashimoto for sharing KD3 skeletal muscle cells, David Sibley for sharing DG52 antibodies, Wesley Van Voorhis and Wolfgang Bohne for sharing BKIs and HDQ, respectively; Alexander Neumann for sharing CCEAN software; Gereon Schares, Marie-France Cesbron-Delauw, Jonathan Howard, Boris Striepen and Dominique Soldati-Favre for *T. gondii* strains and Anton Aebischer and Dominique Soldati-Favre for helpful discussions and Aura María Bastidas Quintero for technical assistance M.B., C.C., D.M., E.H., J.S., F.M. are funded by the Federal Ministry of Education and Research (BMBF) under project number 01KI1715 as part of the "Research Network Zoonotic Infectious Diseases". F.S. is a senior member of graduate schools GRK 2046 and IRTG 2290, supported by the German Research Council (DFG). T.S. receives support from the Deutsche Forschungsgemeinschaft (DFG) (STE 2348/2), M.M.L. is funded (Research Grants-Doctoral Programs in Germany) by the German Academic Exchange Service (DAAD). M.B., C.C., D.M., E.H., J.S., F.M., T.H. and F.S. receive internal support from the Robert Koch-Institute.

## Author contributions

Conceptualization: M.B.; C.C.; Methodology: M.B., F.S., C.C., D.M., M.M.L., T.S.; Investigation: C.C., D.M., E.H., M.M.L., T.H., J.S., F.M.; Writing—original draft: M.B. and C.C.; Writing—review & editing: all authors; Funding acquisition: M.B.; Supervision: M.B., F.S., T.S.

## Funding

## Competing interests

The authors declare no competing interests.
