## [Peer Review File · Nature Communications]

In vitro maturation of *Toxoplasma gondii* bradyzoites in human myotubes and their metabolomic characterizationReviewers' Comments:

Reviewer #1:

Remarks to the Author:

Christiansen et al. have developed a novel in vitro model of *Toxoplasma gondii* infection that allows for the study of mature tissue cysts. Current methods limit the study of mature cysts to murine infection, which prevents analyses that require high levels of sample input, such as metabolomics. Here, the authors demonstrate that human myotubes can support development of mature *T. gondii* tissue cysts. The mature cysts are resistant to treatments with pepsin and anti-parasitic compounds—traits that are considered diagnostic for such stages. Moreover, these cysts are orally infectious in mice. Using this new model, the authors characterize the metabolomes of tachyzoites and bradyzoites to identify metabolic differences between life stages of the parasite, and provide evidence that bradyzoites do not require a functional TCA cycle. The authors demonstrate that their myotube-based system will be a useful model for the parasitology community, as it is the first in vitro model that sustains long-term *Toxoplasma* cyst development.

Overall, the effectiveness of the system is clear, but more data and discussion are required in several places. Given that murine oral infectivity is the standard definition of functionally mature cysts, the authors should provide more robust data from mouse infection experiments. In addition, more justification for several methods and conclusions is required, such as the formulation of the resistance score and, in particular, the claim that the TCA cycle is dispensable in bradyzoites.

MAJOR CONCERNS

1. In Figure 2C, the data is displayed as % of DBA-positive cysts that are also SAG1 negative. The authors should also provide data showing the percentage of all vacuoles that are DBA positive to provide a broader picture of the proportion of tachyzoites vs. bradyzoites in the differentiating populations, and not just the number of mature cysts. This data could at least be provided as a supplement.
2. In the resistance score, it is unclear why the score is corrected using the number of days observed. If correctly interpreted, longer periods of observation would diminish the difference between treatments. The authors should provide clearer justification for the use of this term in the calculation and an explanation of this score in the text.
3. If the benchmark of cyst maturity is that they are orally infectious to mice, the data shown from mouse experiments should be more robust. For example, in Figure 3, murine infection should also be performed with tachyzoites derived from myotube cultures to ensure that seroconversion and oral infectivity is reflective of bradyzoites specifically, and not other features of the myotube-derived culture. The weight loss displayed in Figure 3F also does not seem very significant compared to data for *Toxoplasma* infection in the literature. Finally, in Figure S3 it would be more useful to provide a quantitative metric, such as counts of cysts isolated from brains of infected mice, instead of a representative cyst.
4. The authors do not sufficiently address the drawbacks to the metabolomics approach undertaken, such as measuring metabolite levels rather than metabolic flux. This could be addressed by considering caveats and alternative explanations in the text.
5. How do the authors know cysts are permeable to NaFac? This should be discussed in the text. If permeability cannot be verified, the claim that the TCA cycle is “dispensable” in bradyzoites is not well-supported beyond the observational metabolomics data.

MINOR CONCERNS

1. Although comparison to HFFs is discussed in the text, the authors should consider providing data

for time to differentiation in HFFs, such as is shown in Figure 2C for myotubes. This direct comparison would be a helpful baseline to evaluate the improvements the myotube model provides.

2. In Figure S2, images should have a tachyzoite counter-stain so that the reader can easily assess the proportion of bradyzoites in the total population.

3. Figure 3B: it is difficult to make out the treated tachyzoite line, and distinguish the three dashed lines (including the limit of detection).

4. Representative images of wells from the growth experiment in Figure 3B could help the reader more easily interpret this data.

5. In figure 5E, the data would be easier to interpret if there were a common denominator on both the stage-dependent and host-dependent sides of the figure. Displaying the heatmap as cysts/Tz (myotubes) and Tz (HFFs)/Tz (myotubes) would allow the reader to more easily compare the data.

Reviewer #3:

Remarks to the Author:

In their manuscript, Christiansen et al present a myotube-based system for generating mature *Toxoplasma* cysts, which they used subsequently for testing the effects of drugs on parasite viability or to perform metabolomic analyses.

The authors managed to standardize an in vitro culture system for generating mature cysts, which is a major accomplishment. Others had previously described spontaneous stage conversion in myotubes, but the yield and reproducibility of the method was not optimal. Here, thorough characterization of the cysts obtained in vitro with the present method clearly shows that they display the typical features of mature cysts. Importantly, they can also be used in an assay developed by the authors to assess the sensitivity of bradyzoites to antiparasitic compounds. Being able to screen compounds on in vitro-obtained cysts exhibiting the properties of mature cysts, is a very significant achievement that will be really useful to the community, as drugs able to target the chronic stage are direly needed.

The authors also use their model of in vitro-generated cysts to provide some indication that the TCA cycle is not needed for mature bradyzoite survival (something that was already suggested by the existing literature). However, this part of the manuscript is supported by a limited number of analyses (metabolites identified or drugs used), thus I think these findings may be strengthened by further analyses. So overall, I am clearly positive on the first part of this manuscript, which describes a robust and useful method for generating mature in vitro cyst, offering interesting perspectives for discovering novel anti-parasitic drugs. On the other hand, I am less convinced about the biological insights provided by the second part of the paper.

Main points

One important limitation for accurate metabolomics analysis of in vitro-generated bradyzoites is the sample preparation process. Many important cellular metabolites can quickly be metabolized by enzymes or degraded when exposed to changes in temperature or external medium. Consequently, the level of these metabolites may vary very rapidly during sample preparation, and that would change their final concentrations, producing results that may not accurately reflect the population's true metabolic state. This is why a rapid quenching of the metabolism is needed and usually achieved by rapid freezing of the samples. Here, bradyzoites were left for at least an hour in ice-cold PBS prior to metabolite extraction. This is not only unlikely to stop metabolic activity, but clearly also creates starvation-like conditions. First, these limitations inherent to bradyzoite preparation should be clearly

mentioned in the manuscript. Second, it is currently unclear if the control tachyzoites were treated in the same conditions (left for an extended period of time in PBS) and this should be clarified as it could have a significant impact on metabolite levels. Finally, how the host cell extracts were prepared for background subtraction should be clarified (I believe host cells were scrapped and syringed, then the lysate was incubated with the DBA column and processed as with the cyst samples).

Perhaps as a consequence of this, only a couple of metabolites of the TCA cycle were detected and quantified. The GABA shunt is linked to the TCA cycle, but GABA is not directly a TCA cycle intermediate, although it can feed into the cycle (but this is the case for several amino acids too). Thus these variations may not be enough to fully support a down-regulation of the cycle. The existing bibliography already suggested a non-functional TCA cycle in bradyzoites and conversely an increased glycolytic activity, and this was based on enzymatic assays (Denton et al 1996). Specific enzymatic assays could thus be performed on the in vitro-obtained mature cysts described in the present paper to strengthen the metabolomics findings. Quantitative proteomic or transcriptomic analyses showing specific lower expression of TCA cycle-related enzymes in the same samples of mature bradyzoites may also potentially strengthen this hypothesis. Transcripts of the key enzymes in the TCA cycle were for example already shown to be less abundant in chronic infection compared to acute (see for example PMID: 25240600).

The resistance of mature cysts to sodium fluoroacetate suggests the TCA cycle is indeed dispensable in mature bradyzoites, but as it is not known if this particular drug can reach its target efficiently in the cysts, the parallel use of another drug targeting the TCA cycle would reinforce this finding. Generating a conditional TCA cycle-specific mutant in bradyzoites would be even better (but I do realize this would involve substantial additional experimental efforts).

Specific points

The myogenic index used to quantify myoblast fusion is defined not just as "the fraction of total nuclei that reside in multinucleated cells" (l. 359), but precisely as the number of nuclei residing in cells containing three or more nuclei, divided by the total number of nuclei.

l. 574-582: As the TCA cycle generates the reducing equivalents which are required to transfer electrons to the mitochondrial electron transport chain, and the authors suggest the TCA cycle is not fully functional in bradyzoites, there is an apparent discrepancy in considering the bc1-complex as a relevant drug target in chronic stages. Some in vivo studies have shown atovaquone and quinolones may reduce the cyst burden, so the in vitro system resistance score developed by the authors seemed ideal to solve this apparent discrepancy. Unfortunately, atovaquone could not be tested here and I am not sure the authors can conclude on in vitro bradyzoite resistance to HDQ. To my knowledge, HDQ has not been assessed on bradyzoites before, and I know it interferes with the mitochondrial membrane potential of tachyzoites in the sub-micromolar range, but perhaps the 0.1 μM concentration used here is not sufficient. Have the authors tried increasing the HDQ concentration? Can they verify there is an impact on the mitochondrial membrane potential of bradyzoites (which is likely already significantly lower than the one of tachyzoites) at 0.1 μM ?

Fig. 1A shows that a considerable proportion of myoblasts does not differentiate in myotubes. I suppose tachyzoites do invade these myotublasts and can multiply therein, and thus can remain for some time in the culture. Yet, the authors state that "all cultures were DBA-positive and remained stable without being overgrown by tachyzoites for 21 days" (l. 378-379). This is definitely a good thing, but it is not clear how the potential problem caused by tachyzoite growth in myoblasts is alleviated (especially under CO₂-replete conditions): are they removed by regular change of medium?

l. 658: the (Neumann, 2014) reference is missing from the list.

Fig. 2B. Many different strains are cited in the legend, yet it is unclear which ones are those actually shown on the pictures.

Fig. 3. These *in vivo* assays unambiguously show that the *in vitro*-generated cysts can be orally infectious to mice. However, it would also be quite informative to see how the *in vitro*-generated cysts generated in this study would compare with 50 or 500 brain-derived cysts inoculated by gavage. This would help assessing better how infectious the *in vitro*-generated cysts are: for example, with 50 *in vitro*-generated cysts there seems to be a relatively modest weight loss, if any (I do not think the authors can even state there is a “transient” weight loss -1.429-, as it is not statistically significant).

Fig. 5A. Tachyzoites grown in HFFs seem to cluster into two groups of four datapoints after PCA. Do these correspond to technical replicates in each individual experiments? If that is the case, it shows some degree of variability between independent biological replicates for these particular dataset and maybe this could be commented further.

Reviewer #4:

Remarks to the Author:

This study investigates the metabolism of *Toxoplasma gondii* bradyzoites using a new *in vitro* host infection system. *T. gondii* bradyzoites are responsible for perpetuating chronic infections in a significant proportion of the human population, but the absence of robust experimental systems for generating bradyzoite cysts for biochemical analyses have hampered attempts to identify drugs that target this stage. Previous studies by other groups have shown that *T. gondii* tachyzoites differentiate to bradyzoites with high efficiency in mature skeletal muscle cells (myotubes). In this study, the authors have refined this *in vitro* infection model and shown that they can stably maintain bradyzoites in muscle myotubes for up to 21 days. After validating that the myotube cysts have many of the properties of tissue cysts, they undertook a comparative LC-MS based metabolomic analysis of purified bradyzoite cysts and tachyzoites. Based on these analyses, the authors conclude that parasite metabolism is affected by both the host cell niche and differentiation, and that the TCA cycle activity was decreased in bradyzoite stages, consistent with the observed resistance of this stage to the inhibitor sodium fluoroacetate. The development of the system is well described and represents a potentially useful system for investigating bradyzoite physiology. However, the metabolomic analyses, as currently presented, are limited, require further controls, and do not conclusively support the conclusions made by the authors. Overall, this represents a promising but preliminary analysis of the metabolic state of bradyzoites.

Major issues

1. While it is acknowledged that the availability of bradyzoite material make the metabolomic studies challenging, the relatively low coverage of total metabolome and more specifically metabolites in pathways of central carbon metabolism, make it difficult to draw any strong conclusions from the data. For example, the heat plots in Fig 5c contain no metabolites from glycolysis and the pentose phosphate pathway, and only two metabolites from the TCA cycle (including GABA makes three). It is not clear if other intermediates in these pathways were detected, but were not significantly changed or were removed because they were also present in ‘host background’ samples. If these metabolites were detected, but their abundance did not change, this needs to be shown. Of more fundamental concern, it is difficult to infer anything about pathway activity or flux from steady-state metabolite levels. The apparent decrease in the three TCA cycle intermediates could equally be interpreted as reflecting increased TCA cycle flux and faster turnover of smaller intermediate pools.

2. A second weakness with the metabolomic analyses is the lack of controls for assessing the impact of the extended bradyzoite purification process on bradyzoite metabolism. While the authors show that addition of beads does not contribute substantial metabolite background, they also need to show that

bradyzoite metabolism has not changed substantially during purification. For example, have they assessed the effect of extending/reducing the time used to affinity purify the cysts on metabolite levels, and/or taken tachyzoites through the same series of incubations and washes as the bradyzoites to assess the impact of these procedures on parasite metabolism ?

3. The authors nicely demonstrate that tachyzoites, but not bradyzoites are susceptible to NaFAC, providing support for their conclusion that bradyzoites are not dependent on the TCA cycle for survival. However, NaFAC treatment leads to a block in the early part of the oxidative TCA cycle and bradyzoites could still use the TCA cycle to catabolize other carbon sources, such as glutamine, that by-passes the NaFAC block. Alternatively, they may utilize the TCA in the same way as tachyzoites, but be intrinsically more resistant to NaF treatment because they have slower metabolic rate. These results would be more compelling if the authors could show that that NaFAC was having some effect on bradyzoite/host cell metabolism and/or that bradyzoites are correspondingly more sensitive to inhibitors of glycolysis.

4. For the reasons outlined above, much of the discussion around the role of amino acid transporters in regulating intracellular amino acid levels in bradyzoites are highly speculative.

Reviewer comments

Reviewer #1 (Remarks to the Author):

Christiansen et al. have developed a novel in vitro model of *Toxoplasma gondii* infection that allows for the study of mature tissue cysts. Current methods limit the study of mature cysts to murine infection, which prevents analyses that require high levels of sample input, such as metabolomics. Here, the authors demonstrate that human myotubes can support development of mature *T. gondii* tissue cysts. The mature cysts are resistant to treatments with pepsin and anti-parasitic compounds—traits that are considered diagnostic for such stages. Moreover, these cysts are orally infectious in mice. Using this new model, the authors characterize the metabolomes of tachyzoites and bradyzoites to identify metabolic differences between life stages of the parasite, and provide evidence that bradyzoites do not require a functional TCA cycle. The authors demonstrate that their myotube-based system will be a useful model for the parasitology community, as it is the first in vitro model that sustains long-term *Toxoplasma* cyst development.

Overall, the effectiveness of the system is clear, but more data and discussion are required in several places. Given that murine oral infectivity is the standard definition of functionally mature cysts, the authors should provide more robust data from mouse infection experiments. In addition, more justification for several methods and conclusions is required, such as the formulation of the resistance score and, in particular, the claim that the TCA cycle is dispensable in bradyzoites.

MAJOR CONCERNS

1. In Figure 2C, the data is displayed as % of DBA-positive cysts that are also SAG1 negative. The authors should also provide data showing the percentage of all vacuoles that are DBA positive to provide a broader picture of the proportion of tachyzoites vs. bradyzoites in the differentiating populations, and not just the number of mature cysts. This data could at least be provided as a supplement.

We thank reviewer 1 for the suggestion to better illustrate the degree of differentiation of our bradyzoite cultures. We now formulate: "Strikingly, under both conditions all observed vacuoles of every tested strain were DBA-positive and remained stable without being overgrown by tachyzoites for 21 days. This was also true for the hard to differentiate *RHΔku80* (Figure S2)" (Line 329-331).

To underpin this, we added supplemental Figure S2 showing representative images of 21-day-old *RHΔku80* bradyzoites maintained in pH 7.4 and pH 8.3. We chose images of the *RHΔKu80* strain since this strain is known to be the most difficult to differentiate into encysted bradyzoites.

2. In the resistance score, it is unclear why the score is corrected using the number of days observed. If correctly interpreted, longer periods of observation would diminish the difference between treatments. The authors should provide clearer justification for the use of this term in the calculation and an explanation of this score in the text.

We agree with reviewer 1 that the resistance score should be clarified as it is key for this manuscript. We divided the score by the number of observed days to reflect the difference between the length of observation periods, which are necessary due to differences in cyst

loads between experiments. For example, the five-day difference between 15 days until regrowth of solvent control and 20 days until regrowth of drug-treated parasites would be weighted slightly higher than the same absolute difference between 20 and 25 days. So, the diminishing differences after longer observation periods, as noticed by reviewer 1, is intentional.

This weighting does not impact the statistical significance of the results. For illustration purposes we calculated a simplified RS without the correction as follows and obtained very similar results that are also statistically significant but just have larger error bars.

$$\text{Resistance score} = \# \text{ days vehicle} / \# \text{ days treatment}$$

Hence, we left the RS unchanged and expanded the explanation of the RS in the methods text: “The resistance score (RS) was calculated as the difference of days passed between the detection of vehicle-treated and treated tachyzoite cultures. This difference was normalized to the number of days of total observation to account for differences in initial cyst loads and the resulting re-growth periods between experiments” (Line 157-160).

3. If the benchmark of cyst maturity is that they are orally infectious to mice, the data shown from mouse experiments should be more robust. For example, in Figure 3, murine infection should also be performed with tachyzoites derived from myotube cultures to ensure that seroconversion and oral infectivity is reflective of bradyzoites specifically, and not other features of the myotube-derived culture. The weight loss displayed in Figure 3F also does not seem very significant compared to data for Toxoplasma infection in the literature. Finally, in Figure S3 it would be more useful to provide a quantitative metric, such as counts of cysts isolated from brains of infected mice, instead of a representative cyst.

We in fact think that oral infectivity to mice is actually a relatively weak benchmark for the following reasons: (1) Our pepsin digestions were done at pH 1.5, whereas a stomach of mice maintains milder conditions between pH 3.0 and pH 4.0 [1]. (2) Infectivity to mice has already been observed five to nine days after differentiation in HFF cells [2], whereas other maturity markers like pepsin and drug resistance developing multiple weeks longer (Figure 5, Figure 3). (3) In addition, orally applied tachyzoites have also shown to lead to seroconversion in few animals [3].

However, the data serves as an important point of comparison to other publications and we agree with reviewer 1 that the data should be more detailed. We repeated the *in vivo*

experiment and added experimental groups. The new groups consisted of (1) myotube-derived tachyzoites, (2) HFF-derived tachyzoites, (3) 50 myotube-derived tissue cysts (4) 500 myotube-derived tissue cysts, (5) 10 *in vivo*-derived tissue cysts. We generated a separate panel (Figure 4) showing the results, including the cyst burden in the brain of these mice and deleted representative images of the resulting *in vivo* cysts in Figure S3. We also added relative weight loss and survival curves for mice (Figure S6A-B). We find seroconversion and cysts in 1 and 2 of ten mice that received tachyzoites from HFFs and myotubes respectively. 8 or 9 of 10 mice seroconverted after receiving 50 and 500 myotube-derived cysts and 4 of 4 animals converted after receiving 10 *in vivo* cysts. Hence to achieve reliable seroconversion 500 *in vitro* cysts are needed. To further facilitate comparability to brain-derived cysts, we complemented these data by cyst size estimations. We measure a mean diameter of around 10 μm were as *in vivo* cysts from brain tissue are reported to grow to 56 μm [4]. Approximating the cysts as spheres we estimate a 100-fold smaller volume of *in vitro* cysts and conclude that bradyzoites from myotube-derived cysts are similar in their infectivity to those from *in vivo* cysts. We modified the method section according to new control groups and analyzes (Line 164-210) and added a respective chapter to the results section (Figure 4, Line 385-405).

[1] McConnell EL, Basit AW, Murdan S. Measurements of rat and mouse gastrointestinal pH, fluid and lymphoid tissue, and implications for in-vivo experiments. *J Pharm Pharmacol*. 2008 Jan;60(1):63-70. doi: 10.1211/jpp.60.1.0008. PMID: 18088506.

[2] Fux B, Nawas J, Khan A, Gill DB, Su C, Sibley LD. *Toxoplasma gondii* strains defective in oral transmission are also defective in developmental stage differentiation. *Infect Immun*. 2007;75(5):2580-2590. doi:10.1128/IAI.00085-07

[3] Dubey JP. Re-examination of resistance of *Toxoplasma gondii* tachyzoites and bradyzoites to pepsin and trypsin digestion. *Parasitology*. 1998 Jan;116 (Pt 1):43-50. doi: 10.1017/s0031182097001935. PMID: 9481773.

[4] Watts E, Zhao Y, Dhara A, Eller B, Patwardhan A, Sinai AP. Novel Approaches Reveal that *Toxoplasma gondii* Bradyzoites within Tissue Cysts Are Dynamic and Replicating Entities *In Vivo*. *mBio*. 2015 Sep 8;6(5): e01155-15. doi: 10.1128/mBio.01155-15. PMID: 26350965; PMCID: PMC4600105.

4. The authors do not sufficiently address the drawbacks to the metabolomics approach undertaken, such as measuring metabolite levels rather than metabolic flux. This could be addressed by considering caveats and alternative explanations in the text.

We thank the reviewer for this question and are now stating the aim of our metabolomics analysis in the results section more clearly: "...the metabolome of bradyzoites and its differences to tachyzoites remains largely unknown. To test whether bradyzoites possess a distinct metabolome, we compared the abundance of metabolites between ME49 parasites in their bradyzoite form with intracellular tachyzoites in both HFF cells and myotubes (Figure 6)" (Line 436-439).

For this first initial investigation of the bradyzoite metabolome, relative abundance measurements already reveal many differences to tachyzoites. Isotope-resolved mass-spectrometry-based measurements will certainly be worthwhile in the future but are not required to illustrate broad differences between tachyzoites and bradyzoites. As shown

below, the hypothesis of a dispensable TCA cycle that we derive from these data has been solidified by additional experiments.

5. How do the authors know cysts are permeable to NaFac? This should be discussed in the text. If permeability cannot be verified, the claim that the TCA cycle is “dispensable” in bradyzoites is not well-supported beyond the observational metabolomics data.

This point was raised by all three reviewers and we agree that impermeability of the cyst wall to NaFac is an alternative explanation of the tolerance towards NaFac. While the cyst wall appears permeable to many imported nutrients and secreted metabolites of larger size than NaFac [8; 9], we sought to measure direct metabolic effects of this inhibitor to show its internalization and inhibitory action of the bradyzoite aconitase. NaFac treatment of tachyzoites leads to drastically increased levels of citrate and aconitate in *T. gondii* tachyzoites [10]. LC / MS analysis of treated bradyzoites confirmed an accumulation of aconitate (Figure 7C, D) by more than 2-fold (we did not detect citric acid in this experiment). This demonstrates that the inhibitor is taken up and blocks the parasite aconitase. Moreover, accumulation of aconitate also indicates that the TCA cycle maintains a degree of activity in this parasite stage. We appended the data to the manuscript in Figure 7C, D and added findings to the result section (Line 478-494).

[8] Lemgruber L, Lupetti P, Martins-Duarte ES, De Souza W, Vommaro RC. The organization of the wall filaments and characterization of the matrix structures of *Toxoplasma gondii* cyst form. *Cell Microbiol.* 2011 Dec;13(12):1920-32. doi: 10.1111/j.1462-5822.2011.01681.x. Epub 2011 Sep 22. PMID: 21899696.

[9] Acquarone M., Ferreira-da-Silva M.d.F., Guimarães E.V., Barbosa H.S. *Toxoplasma gondii* tissue cyst: Cyst wall incorporation activity and matrix cytoskeleton proteins paving the way to nutrient acquisition. In: Akyar I., editor. *Toxoplasmosis*. InTech; London, UK: 2017

[10] MacRae, J.I., Sheiner, L., Nahid, A., Tonkin, C., Striepen, B., and McConville, M.J. (2012). Mitochondrial metabolism of glucose and glutamine is required for intracellular growth of *Toxoplasma gondii*. *Cell host & microbe* 12, 682-692.

MINOR CONCERNS

1. Although comparison to HFFs is discussed in the text, the authors should consider providing data for time to differentiation in HFFs, such as is shown in Figure 2C for myotubes. This direct comparison would be a helpful baseline to evaluate the improvements the myotube model provides.

To address this point, we compared bradyzoite maturation over the course of 21 days in bicarbonate-depleted conditions per DBA and SAG1 antigen expression in HFF, KD3 myoblasts and KD3 myotubes (Figure S3). The Pru-tdTomato parasites at neutral and basic pH lysed HFF and myoblasts after 21 and 14 days, respectively. In myotubes, the fraction of DBA-positive and SAG1-negative cysts increased, whereas there is no increase of this population in the other cell types at neutral and basic pH. We supplemented the manuscript with Figure S3 and added a respective section to the result chapter (Line 338-344).

2. In Figure S2, images should have a tachyzoite counter-stain so that the reader can easily assess the proportion of bradyzoites in the total population.

We agree with reviewer 1 in this point and replaced the former images to show an additional SAG1 stain with a higher magnification 96 h post infection (Figure S4). Respective section describing findings was changed in the result chapter (Line 345-351).

3. Figure 3B: it is difficult to make out the treated tachyzoite line, and distinguish the three dashed lines (including the limit of detection).

We cleared up the graph (Figure 3B) by deleting the detection limit and changing the y-axis unit designation and showing data points of the tachyzoite control measurements.

4. Representative images of wells from the growth experiment in Figure 3B could help the reader more easily interpret this data.

We added the requested images of 7-, 14-, 21- and 28-day-old Pru-tdTomato cysts in Figure S5 and also quantified the diameter of our *in vitro* cysts in Figure 4A.

5. In figure 5E, the data would be easier to interpret if there were a common denominator on both the stage-dependent and host-dependent sides of the figure. Displaying the heatmap as cysts/Tz (myotubes) and Tz (HFFs)/Tz (myotubes) would allow the reader to more easily compare the data.

As suggested, we adapted the heatmap as cysts/Tz (myotubes) and Tz (HFFs)/Tz (myotubes) (Figure 6E) and also changed the associated volcano plot in figure 6D accordingly. The respective text passage was rewritten according to changed denominators (Line 467-469).

Reviewer #2 (Remarks to the Author):

In their manuscript, Christiansen et al present a myotube-based system for generating mature *Toxoplasma* cysts, which they used subsequently for testing the effects of drugs on parasite viability or to perform metabolomic analyses.

The authors managed to standardize an in vitro culture system for generating mature cysts, which is a major accomplishment. Others had previously described spontaneous stage conversion in myotubes, but the yield and reproducibility of the method was not optimal. Here, thorough characterization of the cysts obtained in vitro with the present method clearly shows that they display the typical features of mature cysts. Importantly, they can also be used in an assay developed by the authors to assess the sensitivity of bradyzoites to antiparasitic compounds. Being able to screen compounds on in vitro-obtained cysts exhibiting the properties of mature cysts, is a very significant achievement that will be really useful to the community, as drugs able to target the chronic stage are direly needed.

The authors also use their model of in vitro-generated cysts to provide some indication that the TCA cycle is not needed for mature bradyzoite survival (something that was already suggested by the existing literature). However, this part of the manuscript is supported by a limited number of analyses (metabolites identified or drugs used), thus I think these findings may be strengthened by further analyses. So overall, I am clearly positive on the first part of this manuscript, which describes a robust and useful method for generating mature in vitro cyst, offering interesting perspectives for discovering novel anti-parasitic drugs. On the other hand, I am less convinced about the biological insights provided by the second part of the paper.

MAJOR CONCERNS

1. One important limitation for accurate metabolomics analysis of in vitro-generated bradyzoites is the sample preparation process. Many important cellular metabolites can quickly be metabolized by enzymes or degraded when exposed to changes in temperature or external medium. Consequently, the level of these metabolites may vary very rapidly during sample preparation, and that would change their final concentrations, producing results that may not accurately reflect the population's true metabolic state. This is why a rapid quenching of the metabolism is needed and usually achieved by rapid freezing of the samples. Here, bradyzoites were left for at least an hour in ice-cold PBS prior to metabolite extraction. This is not only unlikely to stop metabolic activity, but clearly also creates starvation-like conditions. First, these limitations inherent to bradyzoite preparation should be clearly mentioned in the manuscript. Second, it is currently unclear if the control tachyzoites were treated in the same conditions (left for an extended period of time in PBS) and this should be clarified as it could have a significant impact on metabolite levels. Finally, how the host cell extracts were prepared for background subtraction should be clarified (I believe host cells were scrapped and syringed, then the lysate was incubated with the DBA column and processed as with the cyst samples).

Thank you for the thoughtful considerations. We now refer to the quenching procedure in the results section and more clearly describe the controls that we used in the method section (Line 212-232), but also in the results section it is now stated: "We also estimated artefacts

that are introduced by differences in quenching and preparation procedures of tachyzoites and bradyzoites. We supplemented isolated tachyzoites with beads and processed them similarly to bradyzoites. To correct for background metabolites from contaminating host material and magnetic bead preparations, we included uninfected myotubes and bead-only controls into our analysis. These samples were treated identically to bradyzoite samples“ (Line 444-449).

In particular we used free, quenched tachyzoites and treated them as bradyzoite samples. We added beads and incubated them for one hour in PBS and only replaced the magnet purification by centrifugation. These data were shown in the original manuscript as “tachyzoite + beads (myotubes)” and “tachyzoite + beads (HFF)”. In PCA analysis (Figure S7E) these controls were not distinguishable from tachyzoite samples. We also added a description in the methods section on the preparation of tachyzoites: “Tachyzoite samples were generated in T150 dishes by infecting myotubes and HFF cells with $3,2 \times 10^7$ tachyzoites corresponding to an MOI of 3 for 48 h. Medium was replaced by ice-cold PBS and monolayers were scraped and passaged through a 27G needle. Tachyzoites were filter-purified through a 3 μ m filter (Whatman) and PBS-washed by centrifugation (1,200 x g, 10 min, 0 °C) three times“ (Line 225-229).

2. Perhaps as a consequence of this, only a couple of metabolites of the TCA cycle were detected and quantified. The GABA shunt is linked to the TCA cycle, but GABA is not directly a TCA cycle intermediate, although it can feed into the cycle (but this is the case for several amino acids too). Thus these variations may not be enough to fully support a down-regulation of the cycle. The existing bibliography already suggested a non-functional TCA cycle in bradyzoites and conversely an increased glycolytic activity, and this was based on enzymatic assays (Denton et al 1996). Specific enzymatic assays could thus be performed on the in vitro-obtained mature cysts described in the present paper to strengthen the metabolomics findings. Quantitative proteomic or transcriptomic analyses showing specific lower expression of TCA cycle-related enzymes in the same samples of mature bradyzoites may also potentially strengthen this hypothesis. Transcripts of the key enzymes in the TCA cycle were for example already shown to be less abundant in chronic infection compared to acute (see for example PMID: 25240600).

Thank you for these suggestions. We considered a quantitative analysis of the expression of TCA cycle enzymes, but discarded that idea because of two reasons: (1) the presence of transcripts and proteins only indirectly indicates their activity and has partially been done already [11,12]. (2) Denton et al. only suggests increased glycolytic activity (by measuring PFK, PK and LDH activities in lysates) but similar TCA cycle activity (as per isocitric acid dehydrogenase activity) in both parasite stages (The authors are cautious to conclude on succinate dehydrogenase assays due to low sensitivity of the assay).

However, we agree with reviewer 2 that the TCA cycle metabolites are not optimally covered and this is due to detection limits. To address this point and simultaneously also address major issue #3, we performed metabolomic analyses of NaFAc-treated parasites. We find more than 2-fold accumulation of aconitate in bradyzoites and 6-fold accumulation in tachyzoites (Figure 7C, D) after a seven week and three days treatment, respectively. Overall, the metabolic response was not as dramatic in bradyzoites than in tachyzoites, indicating that the TCA cycle is active on a low level and does not influence many other pathways in bradyzoites. We changed the discussion accordingly (Line 516-539).

[11] Fleige, T., Pfaff, N., Gross, U., and Bohne, W. (2008). Localisation of gluconeogenesis and tricarboxylic acid (TCA)-cycle enzymes and first functional analysis of the TCA cycle in *Toxoplasma gondii*. *Int J Parasitol* 38, 1121-1132.

[12] Pittman, K.J., Aliota, M.T., and Knoll, L.J. (2014). Dual transcriptional profiling of mice and *Toxoplasma gondii* during acute and chronic infection. *BMC genomics* 15, 806.

3. The resistance of mature cysts to sodium fluoroacetate suggests the TCA cycle is indeed dispensable in mature bradyzoites, but as it is not known if this particular drug can reach its target efficiently in the cysts, the parallel use of another drug targeting the TCA cycle would reinforce this finding. Generating a conditional TCA cycle-specific mutant in bradyzoites would be even better (but I do realize this would involve substantial additional experimental efforts).

We thank the reviewer for raising this point and would like to refer to our response to point 5 of reviewer 1. In brief, we found that NaFAc treatment of bradyzoites leads to an accumulation of aconitate, indicating that the inhibitor reaches its target in bradyzoites (Figure 7).

MINOR CONCERNS

1. The myogenic index used to quantify myoblast fusion is defined not just as “the fraction of total nuclei that reside in multinucleated cells” (l. 359), but precisely as the number of nuclei residing in cells containing three or more nuclei, divided by the total number of nuclei.

Thank you for pointing out this inaccuracy. We changed the text accordingly: “Accordingly, the myogenic index, which reflects the fraction of nuclei residing in cells containing three or more nuclei, rose to 0.3 (Figure 1B)” (Line 309-310). We also changed the definition in the caption of Figure 1.

2. l. 574-582: As the TCA cycle generates the reducing equivalents which are required to transfer electrons to the mitochondrial electron transport chain, and the authors suggest the TCA cycle is not fully functional in bradyzoites, there is an apparent discrepancy in considering the bc1-complex as a relevant drug target in chronic stages. Some *in vivo* studies have shown atovaquone and quinolones may reduce the cyst burden, so the *in vitro* system resistance score developed by the authors seemed ideal to solve this apparent discrepancy. Unfortunately, atovaquone could not be tested here and I am not sure the authors can conclude on *in vitro* bradyzoite resistance to HDQ. To my knowledge, HDQ has not been assessed on bradyzoites before, and I know it interferes with the mitochondrial membrane potential of tachyzoites in the sub-micromolar range, but perhaps the 0.1 μM concentration used here is not sufficient. Have the authors tried increasing the HDQ concentration? Can they verify there is an impact on the mitochondrial membrane potential of bradyzoites (which is likely already significantly lower than the one of tachyzoites) at 0.1 μM ?

Thank you for pointing this out. We are very aware of this discrepancy and agree that our *in vitro* bradyzoites would be ideal to address this. We performed additional experiments to ensure that *in vitro* bradyzoites are indeed resistant to HDQ by two strategies: (1) by increasing its concentration 10-fold to 1 μM which corresponds to more than 100x of its IC_{50} in tachyzoites and (2) by measuring its effect on the membrane potential in bradyzoites to address its potential exclusion by the cyst wall (Figure S8).

We found that bradyzoites fully resist one-week exposure to 1 μM HDQ. This HDQ exposure also decreases, but not fully collapses the mitochondrial membrane potential as assessed by Mitotracker staining. We conclude from this data that HDQ indeed enters the mitochondria of bradyzoites but that there might be either several other HDQ-independent sources of electrons that maintain the membrane potential or a less active depolarization by ATP synthase due to dampened ATP production in this stage (see <https://www.biorxiv.org/content/10.1101/2021.05.17.444531v2>). The new data is thus consistent with our original interpretation that the TCA cycle is not needed as an (indirect) source of reduction equivalents and ATP. While revising the manuscript we noticed that we incorrectly introduced HDQ as a bc1-complex inhibitor in the original version of the manuscript. We fixed this error and modified our result section (Line 416-417) and expanded our discussion by referring to the correct targets that includes DHOD and potentially alternative NADH-dehydrogenases in *T. gondii* [1, 2] (Line 528-534 and 566-569). Bc1-complex binding appears to be restricted to Plasmodium parasites [3].

[1] Saleh, A., Friesen, J., Baumeister, S., Gross, U., and Bohne, W. (2007).

Growth inhibition of *Toxoplasma gondii* and *Plasmodium falciparum* by nanomolar concentrations of 1-hydroxy-2-dodecyl-4(1H)quinolone, a high-affinity inhibitor of alternative (type II) NADH dehydrogenases. *Antimicrobial agents and chemotherapy* 51, 1217-1222.

[2] Hegewald, J., Gross, U., and Bohne, W. (2013). Identification of dihydroorotate dehydrogenase as a relevant drug target for 1-hydroxyquinolones in *Toxoplasma gondii*. *Mol Biochem Parasitol* 190, 6-15.

[3] Vallieres, C., Fisher, N., Antoine, T., Al-Helal, M., Stocks, P., Berry, N.G., Lawrenson, A.S., Ward, S.A., O'Neill, P.M., Biagini, G.A., et al. (2012). HDQ, a potent inhibitor of *Plasmodium falciparum* proliferation, binds to the quinone reduction site of the cytochrome bc1 complex. *Antimicrobial agents and chemotherapy* 56, 3739-3747.

3. Fig. 1A shows that a considerable proportion of myoblasts does not differentiate in myotubes. I suppose tachyzoites do invade these myotublasts and can multiply therein, and thus can remain for some time in the culture. Yet, the authors state that "all cultures were DBA-positive and remained stable without being overgrown by tachyzoites for 21 days" (l. 378-379). This is definitely a good thing, but it is not clear how the potential problem caused by tachyzoite growth in myoblasts is alleviated (especially under CO₂-replete conditions): are they removed by regular change of medium?

To clarify this point, we amended the methods sections: "Medium was changed every two days and cells were washed once per week with phosphate buffered saline (PBS). The day after infection, infected monolayers were washed with prewarmed PBS to remove not invaded parasites. Medium was changed every second day and bradyzoite cultures were washed with PBS once a week " (Line 84-88). Also, in response to minor issue 1 raised by reviewer 1, we tested the ability of myoblasts to sustain bradyzoite cultures directly and found that they did not. Monolayers were lysed after 14 days in CO₂-deplete conditions at both neutral and basic pH. In these cells (and HFFs) there was always a large fraction of SAG1 positive parasites remaining. We supplemented the manuscript with Figure S3 and added a respective section to the result chapter (Line 338-344).

4. l. 658: the (Neumann, 2014) reference is missing from the list.

We thank reviewer 2 for this attentive remark. We added the Neumann, 2014 reference to the list.

5. Figure 2B. Many different strains are cited in the legend, yet it is unclear which ones are those actually shown on the pictures.

We apologize for this ambiguous phrasing. The caption of Figure 2B now states "Shown are representative images of tachyzoite controls infected with RH $\Delta ku80$ for 24 h (upper panel), 14-day-old intermediate cysts (middle panel) and mature cysts (lower panel) of the NED strain."

6. Fig. 3. These in vivo assays unambiguously show that the in vitro-generated cysts can be orally infectious to mice. However, it would also be quite informative to see how the in vitro-generated cysts generated in this study would compare with 50 or 500 brain-derived cysts inoculated by gavage. This would help assessing better how

infectious the *in vitro*-generated cysts are: for example, with 50 *in vitro*-generated cysts there seems to be a relatively modest weight loss, if any (I do not think the authors can even state there is a “transient” weight loss -l. 429-, as it is not statistically significant).

We would like to refer to our response to major concern 3 of reviewer 1. In brief, we performed additional infection experiments which can be found in Figure 4. A dose of 10 brain-derived cysts is the maximum inoculum that mice can tolerate, before succumbing to acute toxoplasmosis. One out of five mice already needed to be culled at this inoculum. In our additional experiments, we found 500 *in vitro* cysts to be roughly equivalent to 10 *in vivo* cysts when looking at seroconversion and cyst-load in the brain. We also estimated cyst volume by measuring diameters and found that brain-derived cysts have an approximately 100-fold larger volume than *in vivo* cysts. We modified the method section (Line 165-210) and added a respective chapter to the results section (Figure 4, Line 385-405).

7. Fig. 5A. Tachyzoites grown in HFFs seem to cluster into two groups of four datapoints after PCA. Do these correspond to technical replicates in each individual experiments? If that is the case, it shows some degree of variability between independent biological replicates for these particular dataset and maybe this could be commented further.

Thank you, this observation is well received. The data generally represent two independent experiments of each four replicates. The two clusters of tachyzoites from HFF cells form two apparent clusters that are however, both separated from myoblast-derived tachyzoites and from bradyzoites. We think the reason for these two HFF-tachyzoites clusters are differences in host metabolite contamination because these clusters manifest themselves after subtraction of host cell background, as can be seen in PCA plots in Figure S7D. However, both experiments adhere to the same purification and extraction protocol involving syringe-release of intracellular tachyzoites and filtration. We added to the results section: “The two clusters of HFF-derived tachyzoites represent two independent experiments whose variation manifests itself after host-background subtraction (Figure S6D)” (Line 457-459).

Reviewer #3 (Remarks to the Author):

This study investigates the metabolism of *Toxoplasma gondii* bradyzoites using a new in vitro host infection system. *T. gondii* bradyzoites are responsible for perpetuating chronic infections in a significant proportion of the human population, but the absence of robust experimental systems for generating bradyzoite cysts for biochemical analyses have hampered attempts to identify drugs that target this stage. Previous studies by other groups have shown that *T. gondii* tachyzoites differentiate to bradyzoites with high efficiency in mature skeletal muscle cells (myotubes). In this study, the authors have refined this in vitro infection model and shown that they can stably maintain bradyzoites in muscle myotubes for up to 21 days. After validating that the myotube cysts have many of the properties of tissues cysts, they undertook a comparative LC-MS based metabolomic analysis of purified bradyzoites cysts and tachyzoites. Based on these analyses, the authors conclude that parasite metabolism is affected by both the host cell niche and differentiation, and that the TCA cycle activity was decreased in bradyzoites stages, consistent with the observed resistance of this stage to the inhibitor sodium fluoroacetate. The development of the system is well described and represents a potentially useful system for investigating bradyzoite physiology. However, the metabolomic analyses, as currently presented, are limited, require further controls, and do not conclusively support the conclusions made by the authors. Overall, this represents a promising but preliminary analysis of the metabolic state of bradyzoites.

MAJOR CONCERNS

1. While it is acknowledged that the availability of bradyzoite material make the metabolomic studies challenging, the relatively low coverage of total metabolome and more specifically metabolites in pathways of central carbon metabolism, make it difficult to draw any strong conclusions from the data. For example, the heat plots in Fig 5c contain no metabolites from glycolysis and the pentose phosphate pathway, and only two metabolites from the TCA cycle (including GABA makes three). It is not clear if other intermediates in these pathways were detected, but were not significantly changed or were removed because they were also present in 'host background' samples. If these metabolites were detected, but their abundance did not change, this needs to be shown. Of more fundamental concern, it is difficult to infer anything about pathway activity or flux from steady-state metabolite levels. The apparent decrease in the three TCA cycle intermediates could equally be interpreted as reflecting increased TCA cycle flux and faster turnover of smaller intermediate pools.

We appreciate this criticism and want to point the reviewer to Table S1 and the additional Table S2 in the revised manuscript for detailed information on which metabolites were detected, subtracted as background and whether they are significantly regulated between the sample groups.

We entirely agree on the concern of concluding on pathway activity based on metabolite abundances. Our goal here was to find differences between tachyzoites and bradyzoites that can manifest in changes in relative abundance. We now state this more explicitly: "...the metabolome of bradyzoites and its differences to tachyzoites remains largely unknown. To test whether bradyzoites possess a distinct metabolome, we compared the abundance of

metabolites between ME49 parasites in their bradyzoite form with intracellular tachyzoites in both HFF cells and myotubes (Figure 6)" (Line 436-439).

We indeed found those differences and used them to form a verifiable hypothesis regarding the TCA cycle activity. We added additional data that shows the impact of the aconitase inhibitor on the metabolome of both tachyzoites and bradyzoites (Figure 7C, D). In brief, this inhibitor lead to an accumulation of aconitate in both parasite stages, supporting our initial hypothesis of a less active and also non-essential TCA cycle. Please also refer to our responses to similar concerns of reviewer 1 major concerns 4+5 and reviewer 2 major concerns.

2. A second weakness with the metabolomic analyses is the lack of controls for assessing the impact of the extended bradyzoite purification process on bradyzoite metabolism. While the authors show that addition of beads does not contribute substantial metabolite background, they also need to show that bradyzoite metabolism has not changed substantially during purification. For example, have they assessed the effect of extending/reducing the time used to affinity purify the cysts on metabolite levels, and/or taken tachyzoites through the same series of incubations and washes as the bradyzoites to assess the impact of these procedures on parasite metabolism ?

We appreciate this comment and describe our tachyzoite controls now more explicitly in the results section: "We also estimate artefacts that are introduced by differences in quenching and preparation procedures that are applied to tachyzoites and bradyzoites we supplemented isolated tachyzoites with beads and processed them similarly to bradyzoites." We also estimated artefacts that are introduced by differences in quenching and preparation procedures of tachyzoites and bradyzoites. We supplemented isolated tachyzoites with beads and processed them similarly to bradyzoites. To correct for background metabolites from contaminating host material and magnetic bead preparations, we included uninfected myotubes and bead-only controls into our analysis. These samples were treated identically to bradyzoite samples" (Line 444-449).

We have indeed purified tachyzoite in the same way as our cyst samples and only replaced the affinity-purification step with centrifugation as now mentioned in methods section: "Bead-supplemented tachyzoite controls were processed equally to cyst samples, replacing washing steps via magnetic stand by centrifugation (1,200 x g, 10 min, 0 °C)" (Line 231-233). As shown in Figure S7E, these controls (eight replicates from two experiments using HFF and myotube grown tachyzoites) were indistinguishable by PC analysis and therefore omitted from further analysis. Please also refer to major concern 1 of reviewer 2 addressing a similar point.

3. The authors nicely demonstrate that tachyzoites, but not bradyzoites are susceptible to NaFAC, providing support for their conclusion that bradyzoites are not dependent on the TCA cycle for survival. However, NaFAC treatment leads to a block in the early part of the oxidative TCA cycle and bradyzoites could still use the TCA cycle to catabolize other carbon sources, such as glutamine, that by-passes the NaFAC block. Alternatively, they may utilize the TCA in the same way as tachyzoites, but be intrinsically more resistant to NaF treatment because they have slower metabolic rate. These results would be more compelling if the authors could show that that NaFAC was having some effect on bradyzoite/host cell metabolism and/or that bradyzoites are correspondingly more sensitive to inhibitors of glycolysis.

Thank you for this comment. We like to refer to our answers to similar points raised by reviewer 1 and 2. In brief, we now show metabolic effects of NaFAc on bradyzoites that showed inhibition of the aconitase and further support an active but dispensable TCA cycle in bradyzoites (Figure 7). We appended the discussion by explicitly mention the possibility of continued operation of parts of the TCA cycle on anaplerotic substrates: "It is noteworthy that our data do not exclude the possibility that parts of the TCA cycle remain operational and process anaplerotic substrates, such as glutamine to generate NADH for subsequent ATP production through the mitochondrial electron transport chain (mETC)" (Line 525-527). We also already discussed literature on hexokinase and lactate dehydrogenase mutants that suggest an increased importance of glycolytic ATP production in bradyzoites.

4. For the reasons outlined above, much of the discussion around the role of amino acid transporters in regulating intracellular amino acid levels in bradyzoites are highly speculative.

We re-wrote this part of the discussion citing additional literature in line 540-552 and are more conservative with our assertions. We also addressed concerns regarding the metabolic data above.

Reviewers' Comments:

Reviewer #1:

Remarks to the Author:

The authors have adequately addressed prior concerns and provide significantly more data to support the main points of the study. I remain convinced the methods described will be important for the field and I look forward to the publication of the manuscript.

MAJOR CONCERNS

1. I had previously requested authors show scoring for both SAG1-negative and DBA-positive vacuoles to better characterize differentiation. They now present supplementary data to this end in Fig. S3 for PRU. It is somewhat unfortunate that the data is not presented for the other strains, as this would be an interesting point of comparison. It is curious that so many of the DBL-positive vacuoles are SAG1-positive and this rate remains high at 21 days in the HFFs. This may be another argument for the maturity achieved in the myotubes, which the authors could mention.
2. I am satisfied with the revised explanation of the RS score and its use in the manuscript.
3. The data for oral infectivity is much more robust and emphasizes the important contribution described in this manuscript.
4. The authors now adequately provide caveats for their metabolomics experiments.
5. Measurements of aconitate provide reasonable evidence that the cysts are indeed experiencing relevant concentrations of NaFAC, which significantly strengthens the conclusions of the manuscript.

MINOR COMMENT

Line 54: 'persisting' should be 'persistent'

Reviewer #2:

Remarks to the Author:

Christiansen et al. Present a revised version of their manuscript that includes additional experiments and attempts to clarify some of the points raised by the reviewers. I was initially very enthusiastic about this paper, which I think describes a much-needed approach to generate mature bradyzoites in vitro. However, in spite of some efforts, I think the revised version falls short of providing satisfactory replies to all issues that were identified in the initial manuscript.

Although some experimental details remain to be specified (see below), I think the authors have convincingly shown they can generate cysts in vitro that bear most of the hallmarks of in vivo-derived mature tissue cysts. This is a major accomplishment and will be useful to the Toxoplasma research community. Myotube-derived bradyzoites offer very interesting perspectives for drug screening for instance.

On the other hand, I am much less convinced by the biological insights inferred from the metabolomics analysis of these parasites. Due to the inability to rapidly quench the parasites at low temperature, the protocol to purify bradyzoite is not adapted to comprehensive and high resolution metabolomics studies. This is not critically discussed in the manuscript. The finding that bradyzoites rely less on the TCA cycle, which is clearly put forward in the manuscript and its title, is not strongly supported by metabolomics, which detect too few TCA intermediates. The drug-based approach involving NaFAC, potent inhibitor of the TCA cycle enzyme aconitase, is encouraging as it shows an accumulation of aconitate. However, the accumulation is modest: two-fold in bradyzoite (which might be due to a low TCA cycle activity of bradyzoites indeed), and only slightly more in tachyzoites. It is a

far cry from the 160-fold increase in citrate and aconitate levels MacRae et al detected previously in metabolomics measurements on tachyzoites treated with the same drug (MacRae et al Cell Host Microbe. 2012). In fact, the impact on other TCA metabolites like citrate or isocitrate could not be determined here as they were not detected in the analysis. To me, this shows the limitations of the method. Given the important bradyzoite yield they can achieve, I do not understand why the authors did not try to perform enzymatic assays on myotube-derived bradyzoites. I know previously published transcriptomics data already points to a reduced expression in TCA cycle enzymes, hinting the pathway is likely less active, but it is not the same as assessing it at a biochemical level, which would be really complementary to the metabolomic analysis.

Minor points.

I. 369-370: "robust in vitro culture systems supporting long term maturation of tissue cysts in these natural host cells are lacking" Of note, a recent publication describing an optimized bradyzoite differentiation system in neurons has recently been published and may be mentioned here (Mouveaux T et al. Open Biol. 2021).

Fig. 2, S2, S3: low conversion rates and/or host cell lysis by type I parasites may also depend on the inoculum. Were the same amount of parasites inoculated for each cell line? If a much smaller load of type I parasites is inoculated, can the culture be maintained for a longer period and would the parasites be able to generate tissue cysts?

For the metabolomics studies, it is essential to compare bradyzoites samples with tachyzoites samples processed EXACTLY in the same way. I do not understand why the magnet purification step was replaced by a centrifugation in the case of the control.

I. 406-407: does not correspond to the figures, S3A is for DBA positive, S3B is for SAG1 negative.

I.422-423: "develop functional hallmarks of in vivo cysts, such as resistance to temperature stress and pepsin digestion", cysts are not resistant to pepsin per se, it is even used to have them release bradyzoites.

Figures S8 and S9 are not properly referred to in the paper.

Reviewer #3:

Remarks to the Author:

The authors have addressed several of the issues raised by the reviewers in this revised version of the manuscript. In particular, they have included new data comparing the oral infectivity of the MT-derived cysts from the other parasites stages. They have also provided new data showing that NaFAC treatment of MT-cysts leads to ~ 2-fold accumulation of aconitate, which they interpret as evidence that NaFAC is indeed taken up by these stages and that this treatment has resulted in effective inhibition of the initial reactions in the oxidative TCA cycle. While the authors should be commended for extending their analyses to address the latter point – which was raised by all three reviewers – I am not convinced that the data are yet robust enough. The increase in aconitate levels in NaFAC treated MT-cysts is very modest and well within biological variability, particularly given that NaFAC will also affect host cell metabolism, with possible knock-on effects on bradyzoite metabolism. The lack of changes in other metabolites (including apparent lack of measurable accumulation of citrate) adds to the suspicion that the aconitate increase may not be directly related to NaFAC inhibition in bradyzoites. NaFAC treatment leads to the formation of fluoro-citrate (as well as fluoroacetyl-CoA), which inhibits aconitase activity and should be detected by LC-MS. Confirmation that NaFAC leads to the formation of fluoro-intermediates would provide direct evidence that NaFAC is indeed taken up by bradyzoite cyst stages. This information may already be available in existing LC-MS datasets. As noted by the

reviewers, this important issue could also be resolved by labelling bradyzoites with ^{13}C -glucose and would greatly add to the significance of the paper and validation of the system. In the absence of these additional experiments, and direct evidence that the TCA cycle has indeed been blocked in NaFAC-treated MT-cysts, the authors need to further tone back their conclusions that bradyzoites do not rely on the TCA cycle (in the title and through-out text). Other parts of the discussion also remain highly speculative – in particular, the discussion of the significance of decreased Met, Thr, Phe levels in bradyzoites (line 645-655) is conjecture and could be removed.

Response to Reviewers

Reviewer 1

1. I had previously requested authors show scoring for both SAG1-negative and DBA-positive vacuoles to better characterize differentiation. They now present supplementary data to this end in Fig. S3 for PRU. It is somewhat unfortunate that the data is not presented for the other strains, as this would be an interesting point of comparison. It is curious that so many of the DBL-positive vacuoles are SAG1-positive and this rate remains high at 21 days in the HFFs. This may be another argument for the maturity achieved in the myotubes, which the authors could mention.
2. I am satisfied with the revised explanation of the RS score and its use in the manuscript.
3. The data for oral infectivity is much more robust and emphasizes the important contribution described in this manuscript.
4. The authors now adequately provide caveats for their metabolomics experiments.
5. Measurements of aconitate provide reasonable evidence that the cysts are indeed experiencing relevant concentrations of NaFAc, which significantly strengthens the conclusions of the manuscript.

MINOR COMMENT

Line 54: 'persisting' should be 'persistent'

We thank reviewer 1 for the positive comments! As suggested, we changed persisting to persistent.

Reviewer #2 (Remarks to the Author):

Christiansen et al. Present a revised version of their manuscript that includes additional experiments and attempts to clarify some of the points raised by the reviewers. I was initially very enthusiastic about this paper, which I think describes a much-needed approach to generate mature bradyzoites in vitro. However, in spite of some efforts, I think the revised version falls short of providing satisfactory replies to all issues that were identified in the initial manuscript.

Although some experimental details remain to be specified (see below), I think the authors have convincingly shown they can generate cysts in vitro that bear most of the hallmarks of in vivo-derived mature tissue cysts. This is a major accomplishment and will be useful to the Toxoplasma research community. Myotube-derived bradyzoites offer very interesting perspectives for drug screening for instance.

On the other hand, I am much less convinced by the biological insights inferred from the metabolomics analysis of these parasites. Due to the inability to rapidly quench the parasites at low temperature, the protocol to purify bradyzoite is not adapted to comprehensive and high resolution metabolomics studies. This is not critically discussed in the manuscript.

In response to your comment, which we appreciate, we addressed the quenching procedure in the results (*"Bead-supplemented tachyzoite controls that underwent incubation with similar purification procedure as tissue cysts and were incubated with beads for one hour were indistinguishable from pure tachyzoites."*) and discussion along with the differences of the responses to NaFAc in the MacRae et al 2012 paper (*"Extensive accumulation of citric acid and aconitate has previously been shown to occur in extracellular tachyzoites already after four hours of treatment⁶. We attribute the lower accumulation of aconitate we observe in bradyzoites to the extended treatment over seven days and the longer quenching and purification procedure required for intracellular parasites."*). In short, the comparison here is between extracellular tachyzoites and intracellular bradyzoites. Both differ in their amenability to rapid quenching and biology. In our manuscript we explicitly compare the bradyzoite purification process experimentally with the established method to prepare intracellular tachyzoites for metabolomics measurements (Macrae et al., 2012, Blume et al 2015). Both methods lead to very similar metabolomes (Figure S8).

The finding that bradyzoites rely less on the TCA cycle, which is clearly put forward in the manuscript and its title, is not strongly supported by metabolomics, which detect too few TCA intermediates.

We toned down our statement that the TCA cycle is dispensable in bradyzoites throughout the manuscript and discuss the resistance to NaFAc throughout the manuscript instead. Our metabolomics data comparing bradyzoites and tachyzoites as well as comparing treated and untreated bradyzoites however are consistent with a less important role of the TCA cycle. We also explicitly discuss the possibility that a part of the TCA cycle remains operational in presence of NaFAc (*"It is noteworthy that our data do not exclude the possibility that parts of the TCA cycle remain operational and may process anaplerotic substrates, such as glutamine to generate NADH for subsequent ATP production through the mitochondrial electron transport chain (mETC)."*).

The drug-based approach involving NaFAc, potent inhibitor of the TCA cycle enzyme aconitase, is encouraging as it shows an accumulation of aconitate. However, the

accumulation is modest: two-fold in bradyzoite (which might be due to a low TCA cycle activity of bradyzoites indeed), and only slightly more in tachyzoites. It is a far cry from the 160-fold increase in citrate and aconitate levels MacRae et al detected previously in metabolomics measurements on tachyzoites treated with the same drug (MacRae et al Cell Host Microbe. 2012).

As noted above, there are important differences between our experiments and the ones performed by MacRae et al. 2012. Those include parasite stage, inhibitor concentration and treatment duration. While MacRae et al. treated extracellular tachyzoites that are very amenable to quenching, (which are, however, not as physiologically relevant as are intracellular parasites), we used intracellular tachyzoites and bradyzoites that both require more extensive quenching and purification procedures. We also used lower inhibitor concentrations of 1 mM and 0.5 mM instead of 2 mM NaFAc, and a much longer treatment duration of three and seven days vs. only four hours. All three of these factors likely contributed to a lower fold-change.

In fact, the impact on other TCA metabolites like citrate or isocitrate could not be determined here as they were not detected in the analysis. To me, this shows the limitations of the method.

Given the important bradyzoite yield they can achieve, I do not understand why the authors did not try to perform enzymatic assays on myotube-derived bradyzoites. I know previously published transcriptomics data already points to a reduced expression in TCA cycle enzymes, hinting the pathway is likely less active, but it is not the same as assessing it at a biochemical level, which would be really complementary to the metabolomic analysis.

We appreciate this comment, but in our opinion enzyme assays on parasite lysates are well suited to show the presence of enzymes and their activities in the parasite and which is similar to available proteomic and transcriptomic datasets. However, in our opinion those assays do not reflect the importance of implicated pathways or their activity. The later has been shown to depend on small molecules that are likely at unphysiological levels in lysates. For example the pyruvate kinase of *T. gondii* is activated by glucose 6-phosphate and inhibited by glucose 1-phosphate (<https://pubmed.ncbi.nlm.nih.gov/20856875/>) and amylopectin metabolism depends on calcium through a kinase (<https://pubmed.ncbi.nlm.nih.gov/26651943/>). Therefore, we decided that our efforts would best be directed towards the investigation of intact parasites using MS-based methods.

Minor points

1. I. 369-370: "robust in vitro culture systems supporting long term maturation of tissue cysts in these natural host cells are lacking" Of note, a recent publication describing an optimized bradyzoite differentiation system in neurons has recently been published and may be mentioned here (Mouveaux T et al. Open Biol. 2021).

Thank you for pointing to this paper. It shows that P21-positive bradyzoites of a single parasite strain can be raised for two weeks in primary brain cells of mice in a 24-well format. We included this into our introduction.

2. Fig. 2, S2, S3: low conversion rates and/or host cell lysis by type I parasites may also depend on the inoculum. Were the same amount of parasites inoculated for each cell line? If a much smaller load of type I parasites is inoculated, can the culture be maintained for a longer period and would the parasites be able to generate tissue cysts?

Thank you for raising this question. All cultures in Figure 2/S2 were inoculated with MOI 0.1 and there was no host cell lysis limiting the experiment up to 21 days (even with type I strains). We assessed stage conversion for 21 days but cannot extrapolate beyond this period. However, comparing type I parasites after 14 and 21 days of maturation reveals strain and pH-dependent trends. For example, we show that 80 % of GT1 parasites lost SAG1 after 14 days at high pH and that this fraction is not increasing after 21 days. In contrast, 30 % of wild-type RH parasites lost SAG1 expression after 14 days and this fraction rose slightly to above 40 % after 21 days at high pH.

In the experiment shown in Figure S3 we compared the stage conversion of type II PrudTomato parasites in three host cell types and also used MOI of 0.1. Lysis occurred exclusively in myoblasts and HFF cells. We re-plotted this data for increased clarity and also mentioned the MOI used in the materials and methods section ("*For bradyzoite experiments, independent of strain and host cell background, myotube monolayers were infected with 1.3×10^4 extracellular tachyzoites corresponding to an MOI of 0.1 and incubated for the indicated times.*").

3. For the metabolomics studies, it is essential to compare bradyzoites samples with tachyzoites samples processed EXACTLY in the same way. I do not understand why the magnet purification step was replaced by a centrifugation in the case of the control.

Thank you for raising questions about the difference between the preparations of tachyzoite controls and bradyzoite samples for LC / MS measurements. The reason for this control is to estimate the influence of the bradyzoite purification process on the metabolome. We therefore incubated isolated tachyzoites with DBA-coated beads for one hour (Figure 4C). Tachyzoites that underwent this incubation were indistinguishable from the originally isolated tachyzoites in our LC / MS analysis. (Figure S8). Because tachyzoites do not bind to DBA-coated beads magnet purification in exactly the same way was not possible. In our opinion it is unlikely that centrifugation and magnet purification lasting 10-20 min will have a differential

impact on the metabolome of quenched cells, when a one hour incubation does not exert a detectable effect.

4. 406-407: does not correspond to the figures, S3A is for DBA positive, S3B is for SAG1 negative.

Thank you for pointing this out. For a better understanding of the underlying data, we replotted the data for Figure S3A and S3B. Data in both panels correspond to similar experiments done at different pH and include DBA and SAG1 data.

5. 422-423: “develop functional hallmarks of in vivo cysts, such as resistance to temperature stress and pepsin digestion”, cysts are not resistant to pepsin per se, it is even used to have them release bradyzoites.

Thank you very much for pointing to this inaccuracy! We changed the term “tissue cyst” to „bradyzoites“ or “encysted bradyzoites” in the context of pepsin digestion.

6. Figures S8 and S9 are not properly referred to in the paper.

Thank you for noticing this omission. We now added references to this figure (now as Figure S7) in the results section (*“While resistance of bradyzoites to antifolates is well established resistance to mitochondrial inhibitors is less well documented. We sought to test whether exclusion of HDQ by the cyst wall would lead to insensitivity and thus monitored resistance and mitochondrial membrane potential of 28-day old cysts. Up to one 1 μ M of HDQ did not decrease the RS and cyst diameter but decreased the mitochondrial membrane potential estimated by the drop of intensity of Mitotracker staining intensity by 60 % (Figure S7A-D).”*). We refer to Figure S9 in the discussion section.

Reviewer #3 (Remarks to the Author):

The authors have addressed several of the issues raised by the reviewers in this revised version of the manuscript. In particular, they have included new data comparing the oral infectivity of the MT-derived cysts from the other parasites stages. They have also provided new data showing that NaFAc treatment of MT-cysts leads to ~ 2-fold accumulation of aconitate, which they interpret as evidence that NaFAc is indeed taken up by these stages and that this treatment has resulted in effective inhibition of the initial reactions in the oxidative TCA cycle. While the authors should be commended for extending their analyses to address the latter point – which was raised by all three reviewers – I am not convinced that the data are yet robust enough. The increase in aconitate levels in NaFAc treated MT-cysts is very modest and well within biological variability, particularly given that NaFAc will also affect host cell metabolism, with possible knock-on effects on bradyzoite metabolism.

The lack of changes in other metabolites (including apparent lack of measurable accumulation of citrate) adds to the suspicion that the aconitate increase may not be directly related to NaFAc inhibition in bradyzoites. NaFAc treatment leads to the formation of fluoro-citrate (as well as fluoroacetyl-CoA), which inhibits aconitase activity and should be detected by LC-MS. Confirmation that NaFAc leads to the formation of fluoro-intermediates would provide direct evidence that NaFAc is indeed taken up by bradyzoite cyst stages. This information may already be available in existing LC-MS datasets. As noted by the reviewers, this important issue could also be resolved by labelling bradyzoites with ¹³C-glucose and would greatly add to the significance of the paper and validation of the system.

In the absence of these additional experiments, and direct evidence that the TCA cycle has indeed been blocked in NaFAc-treated MT-cysts, the authors need to further tone back their conclusions that bradyzoites do not rely on the TCA cycle (in the title and through-out text). Other parts of the discussion also remain highly speculative – in particular, the discussion of the significance of decreased Met, Thr, Phe levels in bradyzoites (line 645-655) is conjecture and could be removed.

We thank reviewer 3 for these comments. We agree that ¹³C-glucose labeling would be desirable but the associated isotopologue dilution would likely raise the detection limit of many metabolites beyond practical terms. Our chromatography setup is not particularly geared towards detection of dicarboxylic acids (and other negatively charged metabolites) and would likely need to be changed for this experiment. Furthermore, these data might not be very conclusive and may require further control experiments. For example, the uptake of host succinic acid as has been observed in tachyzoites might lead to mixed isotopologue patterns (<https://pubmed.ncbi.nlm.nih.gov/18336823/>).

Thank you also for the suggestions to measure fluoridated metabolites as indicators of NaFAc action. We already searched our existing datasets for respective masses. If those metabolites existed they remain below the detection limit. In absence of authentic chemical standards, we cannot make statements on the occurrence of these metabolites.

We therefore removed the respective part of the discussion about decreased Met, Thr, Phe levels and toned down our conclusions on the importance of the TCA cycle throughout the manuscript including the title. Instead we now discuss the resistance to the aconitase inhibitor.

Reviewers' Comments:

Reviewer #2:

Remarks to the Author:

The authors have toned down their conclusions on the TCA cycle in bradyzoites, which is more reasonable regarding the data they have managed to obtain so far. I have no remaining concern on the technical aspect of the paper and I remain convinced it describes an in vitro differentiation model that will be useful to the Toxoplasma research community.

Reviewer #2 (Remarks to the Author):

The authors have toned down their conclusions on the TCA cycle in bradyzoites, which is more reasonable regarding the data they have managed to obtain so far. I have no remaining concern on the technical aspect of the paper and I remain convinced it describes an in vitro differentiation model that will be useful to the Toxoplasma research community.

We thank the reviewer again for the repeated reviewing of our manuscript.